# Exploiting Vocabulary Frequency Imbalance in Language Model Pre-training

**Woojin Chung**[*]
KAIST[†]
gartland223@gmail.com

**Jeonghoon Kim**[*‡]
NAVER Cloud & KAIST
jeonghoon.samuel@gmail.com

 github.com/Chung-Kim/vocab-imbalance

 huggingface.co/collections/gartland/neurips-2025-vocabulary-frequency-imbalance

## Abstract

Large language models are trained with tokenizers, and the resulting token distribution is highly *imbalanced*: a few words dominate the stream while most occur rarely. Recent practice favors ever-larger vocabularies, but it is unclear where the benefit comes from. To this end, we perform a controlled study that scales the vocabulary of the language model from $24$K to $196$K while holding data, computation, and optimization unchanged. We begin by quantifying the complexity of tokenized text – formalized via Kolmogorov complexity – and show that larger vocabularies reduce this complexity. Above $24$K, every common word is already tokenized as a single token, so enlarging vocabulary only deepens the relative token-frequency imbalance. Word-level loss decomposition shows that larger vocabularies reduce cross-entropy loss almost exclusively by lowering uncertainty on the $2,500$ most frequent words, even though loss on the rare tail rises. Same frequent words cover roughly $75\%$ of tokens in downstream benchmarks, this training advantage transfers intact. We further show that enlarging model parameters with a fixed vocabulary yields the same frequent-word benefit. Our results recast "bigger vocabularies help" as "lowering complexity of tokenized text helps," offering a simple, principled knob for tokenizer–model co-design and clarifying the loss dynamics that govern language model scaling in pre-training.

## 1 Introduction

A language model incorporates a tokenizer that converts a stream of characters into a series of token IDs, each representing a specific substring [22, 47, 12]. Tokenization has re-emerged as an powerful tuning knob for language models, with mounting evidence that simply scaling up vocabulary consistently reduces perplexity and improves downstream accuracy across diverse domains and model scales [58, 65]. Although this trend is consistently observed in practice, the underlying mechanism responsible for it has yet to be thoroughly investigated.

As vocabulary size grows, adding merge candidates segments frequent words in the training data into a single token and pushes infrequent ones deeper into the long tail, sharpening the relative token-frequency distribution [52, 35, 20]. In unigram models, this simply lowers the unigram Shannon entropy of training data toward its optimal loss (i.e., entropy rate) [47]. This intuition does not transfer to language models as they condition their next-token prediction on the preceding context,

---

[*]Equal contribution

[†]Korea Advanced Institute of Science and Technology (KAIST)

[‡]Corresponding author

which contains a mixture of common and rare tokens [2, 8, 42, 7]. Moreover, rare tokens already carry much lower conditional probabilities than marginal ones; mistakes on them incur disproportionately large penalties [45, 39]. Yet rare tokens account for a small share of the entire dataset, making it unclear how a larger vocabulary reallocates capacity between frequent and rare tokens.

In this study, we explore why enlarging vocabulary size improves the performance of language models by expanding vocabulary from 24K to 196K. Viewing BPE [54] with its pre-tokenization rules as a lossless compressor [13, 33], we assess how much the tokenized text is compressed against raw text by an upper-bound on Kolmogorov complexity and first illustrate that expanding the vocabulary reduces this complexity (§3.3). Above 24K, frequent words are already encoded as single tokens, so the primary shift is a heightened imbalance in relative token-frequency distribution (§3.4), regardless of dataset quality (§3.5). Furthermore, a word-level loss decomposition shows that a larger vocabulary reduces the loss of frequent words, thereby lowering the model's overall cross-entropy (§3.6), regardless of dataset and model size.

Analytic experiments trace how enlarging the vocabulary changes both training dynamics and generalization behavior through token-frequency imbalance. Our observation suggests that high frequency words in the pre-training corpus largely coincide with those in downstream benchmarks, both in identity and coverage, explaining the close link between training loss and transfer accuracy (§4.1). Scaling the model itself produces the same benefit: it predicts frequent words more accurately, thereby enhancing overall language-model performance (§4.2).

**Contribution.** We identify that larger vocabularies reduce the complexity of tokenized text, thereby facilitating the model to learn non-i.i.d. patterns in the training data more easily. Our experiments further reveal that beyond a certain size, vocabulary expansion no longer improves segmentation but instead steepens the skewness of the token-frequency distribution. This sharper imbalance alone lowers global cross-entropy by reducing the top $\sim 2{,}500$ frequent words loss despite slight degradation on the rare tail. Through cross-dataset overlap analyses, we demonstrate that exploiting, rather than mitigating, token frequency imbalance causally reduces cross-entropy and boosts downstream accuracy. Finally, we show that parameter scaling replicates the same benefit as vocabulary scaling, both primarily reduce uncertainty on the same set of frequent tokens.

## 2   Motivation

Tokenization – the interface between raw text and the discrete symbols a model actually sees – has resurfaced as a powerful, low-cost lever for improving language model quality [22, 47, 12, 20]. A growing body of evidence finds that simply *increasing the size of the tokenizer vocabulary* yields systematic gains in perplexity and downstream accuracy across domain and model scales [58, 65]. Despite this empirical regularity, the mechanism behind the gain remains underexplored.

The clue may lie in how tokenizers behave as their vocabularies grow. Rajaraman et al. [47] noted that adding merge candidates segments the most frequent words in the corpus into single tokens, making an i.i.d. model over tokens a closer approximation to inherently non-i.i.d. data. Moreover, they empirically confirm that larger vocabularies lower unigram cross-entropy. Simultaneously, this pushes rare vocabularies further into the long tail, yielding a markedly more skewed token-frequency distribution (after the usual whitespace pre-tokenization) [52, 35].

Unfortunately, this explanation does not carry over verbatim to neural language models. In language models, every prediction is conditioned on a variable-length context. Errors on rare vocabularies are disproportionately costly because their conditional probabilities are already orders of magnitude below their marginal frequencies [45, 39], yet they occupy only a tiny fraction of the overall loss. Quantifying how vocabulary growth redistributes modelling capacity between these two regimes, therefore, remains a non-trivial challenge (see the Appendix A for a detailed explanation).

In this work, we tackle the problem head-on by pairing large-scale controlled experiments with analytical diagnostics. We ask:

> *Why does enlarging the tokenizer vocabulary improve Transformer performance,*
> *and which component of the loss benefits most?*

Clarifying this mechanism is essential for principled tokenizer design and for understanding the true drivers of scaling laws in language modelling.

# 3 Experiments

**Guiding Questions.** Before diving into settings and metrics, we spell out the concrete questions that steer our empirical study:

1. **Tokenized Text Complexity**
   Does enlarging the vocabulary reduce the Kolmogorov complexity of tokenized text (§3.3)?
2. **Skew vs. Segmentation**
   Does enlarging the vocabulary *mainly* sharpen the relative token-frequency distribution, or does it still increase single-token coverage of frequent words (§3.4)?
3. **Loss Decomposition**
   When the complexity decreases and frequency skew increases, how is cross-entropy re-allocated between frequent and rare tokens, and which drives the global loss (§3.6)?
4. **Corpus Robustness**
   Are the above effects stable across different data quality—i.e. do high-curation (FineWeb-Edu) and noisier (OpenWebText) corpora exhibit the same trends (§3.5)?

The remainder of this section answers these questions in turn.

## 3.1 Experimental Settings

In these experiments, we train a byte-pair encoding (BPE) tokenizer [54] and estimate token frequencies using a sample of 10 billion GPT-2 tokens from FineWeb-Edu [43] and the entire OpenWebText [15]. For model pre-training, we use approximately 40 billion characters, about 7.5 billion tokens for FineWeb-Edu and 7 billion for OpenWebText with a 49K vocabulary. To compute the metrics below, we also drew an additional 5 billion characters that did not overlap with the training corpus. We report word-level average loss to ensure fair comparison across vocabulary sizes. Whenever a smaller-vocabulary tokenizer splits a word into multiple tokens (i.e., subwords), we sum their individual losses. Our model comprises 85 million non-embedding parameters with pre-layer normalization (pre-LN) [62]. Training uses AdamW [36] ($\beta_1 = 0.9$, $\beta_2 = 0.95$, $\epsilon = 10^{-8}$) with a learning rate of $6 \times 10^{-4}$ that follows a cosine-decay schedule after a 350 million-token warmup, weight decay of $0.1$, and gradient clipping at $1.0$. Every experiment was repeated with five seeds (See Appendix J).

## 3.2 Metrics

**Kolmogorov Complexity of Tokenized Text** Kolmogorov complexity $K(X)$ measures the minimal description length of a bitstring $X$, thus captures its inherent structure and compressibility without assuming any data-generating distribution [29, 16]. This is different from Shannon entropy which is the lower bound of an expected code length for a random variable under its probability distribution (e.g., relative token frequency) [56, 57]. Two concepts are highly correlated each other: shannon entropy approximates an average per-token kolmogorov complexity [18, 40]. We adopt Kolmogorov complexity as a general metric for measuring complexity of tokenized text since different pre-tokenization rules can change token counts dramatically under the same tokenizer settings [35, 52, 48].

Exact Kolmogorov complexity is uncomputable as it reduces to the halting problem [28, 40, 16, 64]. Instead, we calculate a computable upper bound on Kolmogorov complexity, which serves as a practical metric for measuring the complexity of compressed data [16]. As BPE tokenization [54] is derived from a BPE compression algorithm [14], tokenized text can be viewed as compressed data. For a BPE-tokenized bitstring $X^N$, an asymptotic upper bound on $K(X^N)$ is given by

$$K(X^N) \leq N H(p) + V \log_2 N + O(\log N),$$

where $N$ is the total token count of tokenized text, $H(p)$ is the unigram Shannon entropy of the token distribution, and the $V \log_2 N$ term accounts for the prefix-free encoding of the token (e.g., token frequency table). Since each token count is an integer $\leq N$ requiring $\lceil \log_2(N+1) \rceil$ bits, the BPE model is $\leq V \log_2 N$ bits. The first term upper-bounds the tokenized data size, the second bounds the frequency table size, and the third captures minor logarithmic overhead. Since modern language models are trained on a massive corpus with billions of tokens, the $N H(p)$ component dominates. We use $K(X^N) \approx N H(p)$, focusing on the complexity of the tokenized sequence itself. Accordingly, we take $H(p)$ as the unigram Shannon entropy, despite it being a loose upper bound, to enable fair token-count comparisons across tokenizers with different vocabulary sizes.

**Loss Decomposition Metrics** In these experiments, we calculate three metrics to assess the language model's performance both for individual vocabulary types and overall: (1) Total Loss, (2) Average Per-Word Loss, and (3) Global Cross-Entropy Loss.

For each vocabulary $v$, we accumulate its *Total Loss*:

$$\text{Total Loss}(v) \; = \; \sum_{t \in N} \sum_{i=1}^{|t|} \mathbb{1}(v = t_i) \left[ -\ln p(t_i \mid t_{<i}) \right],$$

where $N$ is the set of evaluation documents, $\mathbb{1}(v = t_i)$ is the indicator that the $i$th token equals vocabulary $v$, and $-\ln p(t_i \mid t_{<i})$ is the negative log-likelihood of that token. Total loss measures the sum of negative log-likelihoods for each vocabulary, reflecting its impact on the model's loss [38].

*Average Per-Word Loss* for vocabulary $v$ is defined as

$$\mu(\ell_v) \; = \; \frac{\text{Total Loss}(v)}{T_v(N)},$$

where $T_v(N)$ is the total count of occurrences of $v$ in the training data. This represents the mean negative log-likelihood across all occurrences of each vocabulary $v$ [38].

The *Global Cross-Entropy Loss* is the weighted average of these per-word losses:

$$\text{Global Cross Entropy Loss} = \sum_{v} \frac{T_v(N)}{T(N)} \, \mu(\ell_v),$$

where $T(N) = \sum_v T_v(N)$ is the total token count in the training data. Global Cross-Entropy Loss reflects the model's average uncertainty per prediction [25, 38].

### 3.3 Increasing vocabulary size reduces complexity of tokenized text

Naturally occurring data contains a shared structure and inherently low complexity and models can learn effectively from training data, achieving high accuracy across diverse tasks and generalizing to unseen datasets [16]. To quantify how larger vocabulary size helps the tokenizer approximate natural data's low intrinsic complexity and produces simpler tokenized sequences, we measure the upper bound of Kolmogorov complexity and the normalized compression ratio (NCR). NCR is a practical compressibility metric $\text{NCR}(x; C) \; = \; \frac{|C(x)|}{|x|}$ where $|x|$ is the byte length of data $x$ and $|C(x)|$ is its size after lossless compression by compressor $C$; in our formulation, $K(X^N) \approx |C(x)| = N \, H(p)$.

Table 1: Upper bound of Kolmogorov complexity $(K(X^N))$ and NCR for the 45.97 billion byte FineWeb-Edu corpus with $K(X^N)$ measured in bytes.

| Vocab size | $K(X^N)$ | NCR |
|---|---|---|
| 24K | $10.74B$ | 0.234 |
| 49K | $10.43B$ | 0.227 |
| 98K | $10.23B$ | 0.223 |
| 196K | $10.16B$ | 0.221 |

Table 1 reports the upper bound on Kolmogorov complexity and NCR for the 45.97 billion byte FineWeb-Edu corpus (10B GPT-2 tokens). The results show that BPE tokenizers with larger vocabularies yield lower complexity and NCR by segmenting frequent non-i.i.d. character sequences in text (e.g., words) as a single token. This facilitates models to learn non-i.i.d. statistical patterns in natural text more easily by offloading the low-level pattern learning, thereby simplifying language modeling.

### 3.4 Segmentation already saturates; vocabulary growth mainly sharpens frequency skew

To quantify how each factor evolves with vocabulary growth, we disentangle two factors: (i) relative token-frequency imbalance (Figure 1a) and (ii) segmentation efficiency (Figure 1b)—the fraction of frequent words that are represented by exactly one token. *Frequent words* are operationally defined as the top $2,500$ word types, chosen because this cutoff already covers $\geq 70\%$ of corpus tokens in both FineWeb-Edu (74.4 %) and OpenWebText (75.5 %; full coverage curves in Appendix B).

In this experiment, relative token-frequency imbalance is quantified by the Jensen–Shannon divergence (JSD) from a uniform distribution of the same vocabulary size for a fair comparison across

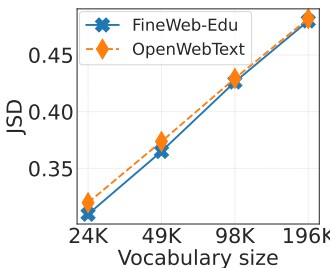 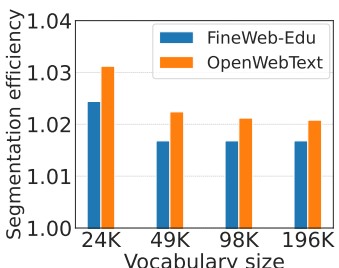 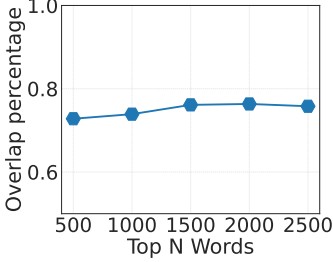

(a) Token-frequency imbalance  (b) Number of tokens per word  (c) Overlap of frequent N words

Figure 1: Figure 1a shows that increasing vocabulary size exacerbates relative token-frequency imbalance. In other words, enlarging the vocabulary size introduces more rare tokens, causing the relative token-frequency distribution to be further from a uniform distribution. Figure 1b reveals that a 24K vocabulary size tokenizer already segments 2,500 frequent words as a single token regardless of dataset quality. This implies that further vocabulary growth offers no added benefit for estimating the probabilities of frequent words. Figure 1c shows that the most frequent $n$ words in fineWeb-Edu and OpenWebText largely overlap, highlighting the universality of frequent vocabulary across different datasets. We report the most frequent 2,500 words in FineWeb-Edu and OpenWebText, which account for approximately 74.4% and 75.5% of each dataset, respectively.

different vocabulary sizes [67, 12, 52]. Segmentation efficiency gauges the average token count per word, that is, how many tokens the tokenizer needs, on average, to encode each of the 2,500 most frequent words. Figure 1a shows that JSD grows monotonically with vocabulary size, reflecting greater token-frequency imbalance. In contrast, Figure 1b indicates that the segmentation efficiency exceeds 95 % by 24K tokens and plateaus later. In other words, beyond 24K, the vocabulary gets larger *without* providing additional single-token coverage for frequent words. Within the widely used $24K - 196K$ range [51, 1, 49], enlarging the vocabulary chiefly amplifies relative token-frequency imbalance rather than improving segmentation efficiency. This finding tempers the common view that "bigger vocabularies help by approximating word-level tokens" [47]: that mechanism appears to saturate once frequent words are already tokenized as a single token.

### 3.5 Skew-driven effects persist across corpora quality levels

To test whether the effects of increasing vocabulary size hold across datasets of varying quality, we conduct experiments on both FineWeb-Edu [43] and OpenWebText [15]. Figure 1a and 1b demonstrate that the impact of increasing vocabulary size has a similar effect on both high-quality datasets (e.g., FineWeb-Edu [43]) and the lower-quality ones (e.g., OpenWebText [15]). Figure 1c further shows that the frequent 2,500 words coincide with nearly 75% each other, highlighting the universality of frequent vocabulary across different corpora. Overall, the findings indicate that vocabulary size expansion produces the same effects irrespective of corpus quality and that high-frequency words substantially overlap across datasets.

### 3.6 Larger vocabularies reduce frequent words uncertainty and global loss

The effect of tokenized text complexity and relative token-frequency imbalance on language models has not yet been explored. Each prediction in a language model depends on previous tokens, and the conditional nature of these probabilities adds significant analytical complexity. However, it is possible to hypothesize that loss of rare token prediction weighs heavily in the overall loss, as conditional probability is often orders of magnitude lower and therefore yields much higher per-token losses [45, 39]. To address this presumption, we empirically examine how reducing loss on frequent word prediction at the expense of rare word predictions affects overall model performance. Models are pre-trained on 40B bytes and evaluated on a separate, non-overlapping 5B byte split of FineWeb-Edu. Figure 2a shows that enlarging the vocabulary steadily lowers the average per-word loss for the frequent words, where the gap between 24K and 196K is roughly 0.1 nats for the frequent 2,500 words. In contrast, the average per-word loss for the rarest 20,000 words rises with vocabulary size, increasing from roughly 11.183 to 13.399 nats. As shown in Figure 2b, the global cross-entropy declines from about 3.179 to 3.136 nats, implying that gains on frequent-word prediction outweigh the degradation on infrequent words. Frequent tokens dominate the objective in all settings: the top

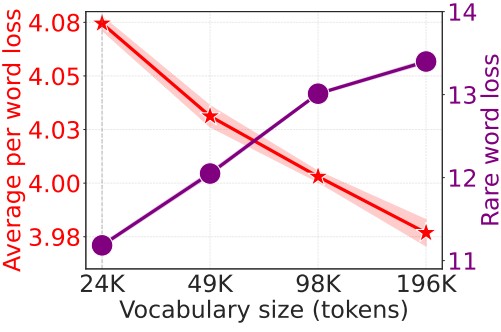
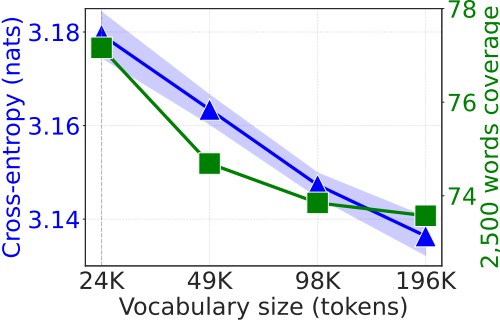

(a) Frequent vs. rare word loss (85M, 10B)    (b) CE-loss & frequent-word coverage (85M, 10B)

Figure 2: Figure 2a illustrates that models with a larger vocabulary size reduce loss on the most frequent 2,500 words while increase loss on the rarest 20,000 words. Nevertheless, Figure 2b shows that the global cross-entropy loss declines as vocabulary size increases, demonstrating that the gains from lower loss on frequent words outweigh the losses from poorer infrequent word estimates. It further reveals that frequent words account for nearly 75% of the total loss, while loss on infrequent words grows with vocabulary size as their conditional probabilities fall due to data sparsity. Models are pre-trained on 40B bytes and evaluated on a disjoint 5B byte split of FineWeb-Edu.

2,500 words account for nearly 75% of total loss. Nevertheless, as the vocabulary expands, losses on infrequent tokens grow, reflecting lower conditional probabilities and reduced predictability for those items. These observations highlight a key takeaway: once loss on frequent words declines, the overall objective is governed by their loss contribution, so further skewness in the token frequency distribution generally improves performance even though it harms rare-token estimates. This finding is consistent with the existing study [47, 38], explaining the benefits of increasing vocabulary size. We observe the same qualitative pattern in the OpenWebText (Appendix C).

### 3.7 Gains from vocabulary expansion hold for larger datasets and models

To assess whether vocabulary size effects persist across data and model scales, we train an 85M (non-embedding) model on a 30B GPT-2 token subset (184.6B bytes) of FineWeb-edu and a 450M (non-embedding) model on a 10B token subset. On the 30B token split, figure 3a shows frequent-word loss drops from 3.845 to 3.742 nats, while rare-word loss rises from 10.199 to 11.787 nats; figure 3b shows global cross-entropy falls from 2.991 to 2.941 nats and frequent-word loss coverage declines from 77.6% to 74.2%, exhibit the same trend reported in Section 3.6. Figure 3c and 3d confirm the same pattern for the 450M models: frequent-word loss decreases from 3.770 to 3.675 nats, rare-word loss increases from 9.694 to 12.762 nats, cross-entropy reduces from 2.989 to 2.888 nats, and frequent word loss coverage from 78.0% to 73.6%. Overall, these findings indicate that vocabulary expansion produces the same effects with larger dataset and model size.

## 4    Analysis

Although our results show that minimizing loss on high frequency tokens is crucial for reducing global cross-entropy, several questions remain unresolved. To pinpoint the causal chain from tokenizer design to model behavior, we now organize our analysis around two additional guiding questions.

**Guiding Questions.**

1. **Transfer Mechanism**
   How does the heavy overlap of the most frequent 2,500 words connect pre-training loss drops to downstream accuracy (§4.1)?

2. **Parameter Count vs. Vocabulary Scaling**
   Can enlarging model size (with a fixed tokenizer) reproduce the same advantage delivered by a larger vocabulary (§4.2)?

The next two subsections answer newly added Q1–Q2 in turn.

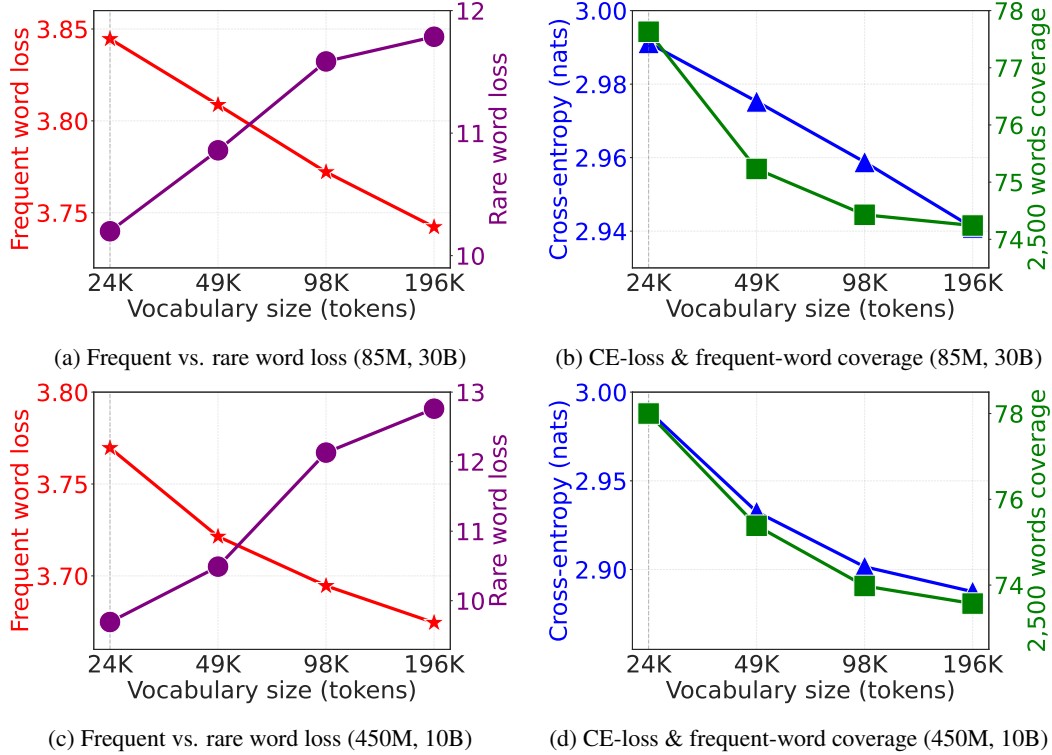

(a) Frequent vs. rare word loss (85M, 30B)  (b) CE-loss & frequent-word coverage (85M, 30B)

(c) Frequent vs. rare word loss (450M, 10B)  (d) CE-loss & frequent-word coverage (450M, 10B)

Figure 3: For an $85M$ model trained on 30B tokens, larger vocabularies reduce the most frequent $2,500$ word loss while increase the rarest $20,000$ word loss; since frequent words dominate, global cross-entropy drops (figure 3a and 3b). 450M model trained on 10B tokens mirrors the pattern (figure 3c and 3d), indicating that these vocabulary-size effects persist across larger datasets and models.

## 4.1 Frequent-word overlap explains downstream performance transfer

Reducing loss for frequent words is key to achieving a lower global cross-entropy loss. But does this effect carry over to downstream task accuracy? Although a strong link between lower loss and better downstream performance has been studied [22, 26], its rationale has not been investigated. To shed light on this phenomenon, we analyze the overlap of frequent words between the pre-training data and evaluation benchmarks and demonstrate that scaling the vocabulary size reduces cross-entropy loss during pre-training and on downstream tasks. Figure 4a shows that the $2,500$ most common words in FineWeb-Edu [43] account for roughly 76-78% of all tokens in ARC [3], HellaSwag [66], and SciQ [61] and about 72% in the PIQA [6] and CC-Main-2023-40 dump [23]. Because lower cross-entropy loss on this CC subset correlates closely with stronger reasoning benchmark scores [23], reducing the frequent words loss and driving down global cross-entropy will improve accuracy across downstream tasks. Consistent with that expectation, figure 4b shows that increasing the vocabulary size from 24K to 196K lowers the average per-word loss on the frequent $2,500$ FineWeb-Edu words by about 0.11 nats, translating into a roughly 0.07 nats drop in global cross-entropy loss on the CC dataset. Figure 4c further confirms that a larger vocabulary also improves average downstream task accuracy of the language model across datasets and model scales. The key insight from this observation is that frequent words highly overlap between the typical training dataset and the downstream benchmark, so the cross-entropy reduction achieved by enlarging the vocabulary size during pre-training naturally translates into better downstream performance.

## 4.2 Parameter scaling recovers the same frequent-word gains

Thus far, we have shown that enlarging the vocabulary size decreases word-level average per-word loss on frequent words, which translates into lower cross-entropy loss and better downstream task performance. One might ask whether similar gains could be achieved without altering the vocabulary size by adjusting other model hyperparameters. Our experiment illustrates that increasing the model's parameter count can replicate the same benefit. To investigate this, we use the Pythia suite, where

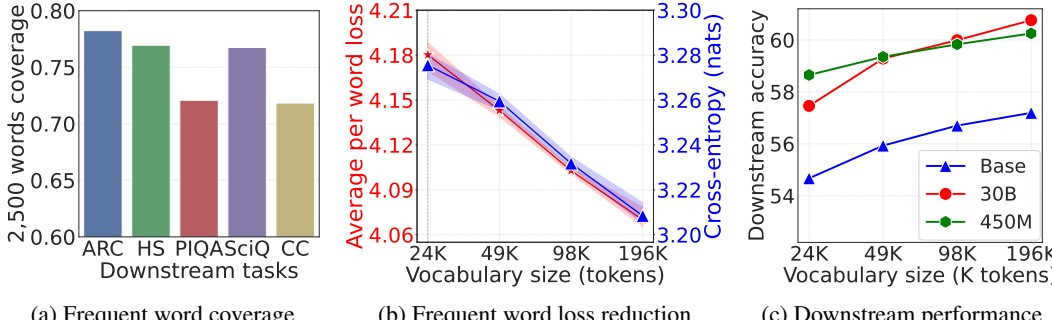

(a) Frequent word coverage     (b) Frequent word loss reduction     (c) Downstream performance

Figure 4: Figure 4a demonstrates that the most frequent $2,500$ words in the FineWeb-Edu comprise nearly $72 - 78\%$ of the tokens in other downstream benchmark datasets as well as the CC-Main-$2023 - 40$ [23]. ARC refers to ARC-Easy, and HS refers to HellaSwag. Figure 4b illustrates that a larger vocabulary reduces average per-word loss on frequent FineWeb-Edu words within the CC dataset, and demonstrates how this translates into lower global cross-entropy loss on CC dataset. Figure 4c confirms that scaling the vocabulary size boosts downstream task performance.

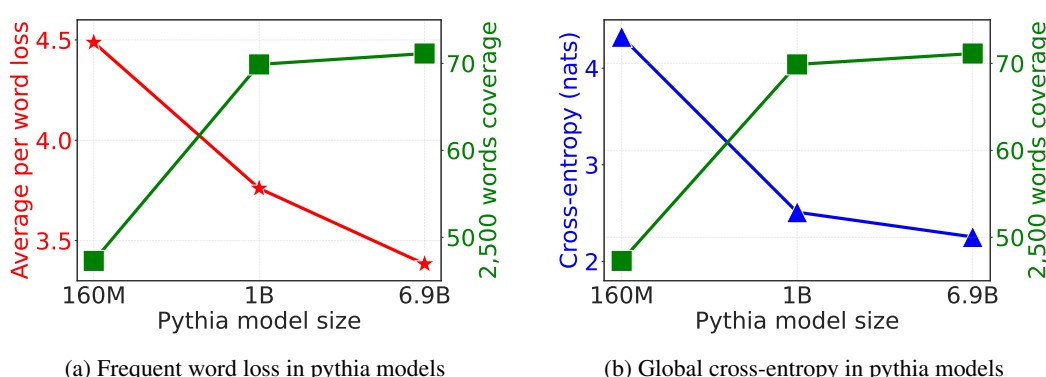

(a) Frequent word loss in pythia models     (b) Global cross-entropy in pythia models

Figure 5: Figure 5a illustrates that increasing model size reduces loss on high frequency words, and the global cross-entropy loss of larger models is overwhelmingly driven by frequent word losses mirroring the effect of increased vocabulary size. However, unlike the pattern in figure 2a, scaling up model size does not exacerbate errors on infrequent tokens. Figure 5b demonstrates that the global cross-entropy loss declines as model size increases, showing the same tendency of scaling up the vocabulary size (figure 2b).

each checkpoint is trained on the same dataset with identical hyperparameters, differing only in model size [4]. We also examine the OLMo-2 series [41] to determine whether the same pattern persists in contemporary large-scale language models (see the Appendix F).

Figure 5 compares Pythia models (160M, 1B, 6.9B parameter count) in terms of word-level average per-word loss, proportion of frequent words losses on total loss, and global cross-entropy loss measured on the Paloma validation dataset [38]. Figure 5a shows that larger models predict frequent words far more accurately: for the most frequent $2,500$ words in FineWeb-Edu, the average per-word loss drops from $4.48$ nats (160M) to $3.76$ nats (1B) and $3.38$ nats (6.9B). Furthermore, frequent words loss dominates the total loss as model size grows, which accounts for almost $70\%$ of the total loss in the Pythia 1B and 6.9B model size, compared with roughly $48\%$ in the Pythia 160M. Unlike the trend observed when only the vocabulary size is increased (Figure 2b), scaling model capacity does not inflate loss on rare tokens; their share of the loss shrinks. Figure 5b illustrates that the global cross-entropy loss on the Paloma validation set drops from $4.32$ nats in the Pythia 160M to $2.51$ nats and $2.26$ nats in the Pythia 1B and 6.9B respectively, mirroring the loss reduction seen when expanding the vocabulary (Figure 8b). From this experiment, we can observe that enlarging the model size also lowers loss on frequent words while effectively ignoring rare token losses, so the global cross-entropy reduction is larger than what scaling up vocabulary size alone can deliver.

# 5 Discussion

## 5.1 Can we reduce the tokenized text complexity without intensifying frequency imbalance?

A larger vocabulary reduces the tokenized text complexity by decreasing the number of tokens and sharpening the token-frequency imbalance. However, Section 3.6 shows that a sharper token-frequency imbalance increases rare-token loss. This raises the question: can we lower tokenized text complexity without enlarging the vocabulary size? SuperBPE [35] is one such example.

SuperBPE employs a two-stage BPE algorithm: in the first stage (up to a threshold vocabulary size $t$), it operates identically to standard BPE, while in the second stage, it abandons whitespace pre-tokenization. By permitting merges across whitespace boundaries, Superbpe directs subsequent merges toward frequent tokens, thereby limiting the introduction of new rare tokens, and preventing further token frequency imbalance as the vocabulary expands.

Table 2: Upper bound of Kolmogorov complexity ($K(X^N)$) and NCR for the $45.97$ billion-byte FineWeb-Edu corpus tokenized with various SuperBPE variants. $t$ and Avg denote the stage-switch vocabulary threshold and an average downstream performance reported by [35], respectively. The SuperBPE variant with the highest average performance exhibits the lowest tokenized text complexity.

| superBPE | $K(X^N)$ | NCR | Avg [35] |
|---|---|---|---|
| 200K($BPE$) | $10.21B$ | 0.222 | 39.8 |
| 200K($t = 180K$) | $10.03B$ | 0.218 | 43.8 |
| 200K($t = 160K$) | $10.05B$ | 0.219 | 43.4 |
| 200K($t = 80K$) | $10.10B$ | 0.220 | 42.9 |

Table 2 reports the upper bound Kolmogorov complexity, NCR of the $45.97$ billion-byte FineWeb-Edu corpus tokenized with various SuperBPE variants, along with average downstream performance reported by [35]. The result shows that the superBPE achieves a lower token count and a less skewed token-frequency distribution than BPE at equal vocabulary. Also, the superBPE variant with the highest average performance exhibits the lowest tokenized text complexity. This finding demonstrates that both vocabulary expansion and the superBPE variants leverage the same benefit: they reduce the complexity of the tokenized text by segmenting common character sequences into single tokens, thereby facilitating the model to learn non-i.i.d. patterns in the data more easily.

## 5.2 Deduplication through the lens of vocabulary frequency imbalance

Deduplication removes exact or near-duplicate content from a corpus, reducing repetition and leakage so that a larger fraction of the dataset is novel [31, 43, 34]. Deduplication can increase the information density of a dataset, i.e., the fraction of non-redundant data in the corpus. For non-i.i.d. data, information density can be proxied by the byte-level entropy rate, the theoretical minimum amount of non-redundant information. However, the entropy rate is only tightly approximable with near infinite context compressors, finite window compressors can distort comparisons before and after deduplication. A more practical loose upper bound is the byte-level bits-per-byte (BPB): under an i.i.d. assumption it quantifies information density, with unigram (byte) entropy upper-bounding the true entropy rate. Under the byte-level vocabulary, greater information density implies a higher BPB, driving entropy toward $\log_2 256 = 8$ bits/byte, which corresponds to a more uniform distribution and lower frequency imbalance. Exploring how deduplication alters BPB, information density, and vocabulary frequency imbalance is a promising avenue for future work.

## 5.3 Why higher token-frequency imbalance degrades machine translation task performance?

Zouhar et al. [67] reported that higher token-frequency imbalance degrades machine translation task performance, contrary to our findings. What explains this contrasting behavior between machine translation and monolingual settings? We presume that this is related to rare word issues in machine translation tasks [55, 53, 27, 37, 19].

In machine translation, the source and target vocabularies overlap minimally: vocabularies that are common in English often occur rarely in French, and vice versa. Conversely, in a monolingual setting, the pre-training corpus and downstream benchmarks share not only vocabularies but also many of the same frequent words, resulting in extensive text overlap. Consequently, BLEU penalizes every missing target-side n-gram, so mistranslating even a handful of infrequent tokens can sharply lower scores. To address this rare-word challenge, methods like vocabulary trimming deliberately shrink the vocabulary size to reduce the impact of low-frequency tokens on translation performance [17, 46, 9].

# 6 Related Work

**Impact of tokenization on transformer language models**    Empirical and theoretical evidence shows that tokenization is a central determinant of both the quality and the speed with which Transformers learn. Early byte or character-level systems such as CANINE [11] and ByT5 [63] avoid sub-word vocabularies but pay a steep price: sequences become an order of magnitude longer, gradients are noisier, and convergence is markedly slower than for sub-word models. Enlarging the BPE dictionary lets even an unigram model approach near-optimal cross-entropy by approximating word-level tokens, whereas the same model without a tokenizer underfits [47]. Furthermore, Controlled scaling studies reveal a log–linear pattern: exponentially expanding the input vocabulary leads to an almost linear drop in loss across model sizes [58]. Huang et al. [22] push this further with "Over-Tokenized Transformers," showing that a 400M parameter model with a 12.8M token encoder matches the loss of a 1B model baseline without extra compute.

Beyond the gains from larger vocabularies, a growing body of work shows that permitting merges across word boundaries (i.e., disabling whitespace pretokenization) improves language model performance. Liu et al. [35], Schmidt et al. [52] turn off white-space pretokenization during BPE tokenizer training once the vocabulary reaches $t$, or at frequency-driven transition points. Taken together, these results indicate that without tokenization, language models stall at character granularity, but with a tokenizer, they converge faster and generalize better.

**Large language model performs a two stage lossless compression**    Large language models perform a two stage lossless compression where tokenizer acts as a pre-compressor and transformer models the tokenized data [13, 33]. Lester et al. [33] further explore the role of tokenizer as a pre-compressor by replacing it with small transformer model with arithmetic coding, and models trained on neurally compressed text underperforms subword tokenizers but beat byte-level LMs. Delétang et al. [13] and Valmeekam et al. [60] has been proposed that transformer language models can be harnessed for compression via arithmetic coding and Heurtel-Depeiges et al. [21] extends this line by probing the possibility of transformers as a universal compressor across text, images, and audio. However, most literatures mixuse the concept of generalization and compression where generalization focuses on predicting unseen data from different datasets by exploiting shared statistical patterns while compression focuses on encoding a fixed dataset in as few bits as possible by modeling its distribution and removing redundancy. This work investigates the role and impact of tokenizer on generalization through the lens of data compression and information theory.

**BPE tokenizer with inductive biases**    BPE tokenizers typically adopt pre-tokenization where regular expressions split text into chunks, sometimes called pre-tokens. Recent work shows that removing pre-tokenization degrades downstream performance across diverse tokenizers [12, 50]. Pre-tokenization rules act as an inductive bias grounded in morphology and grammar, helping the tokenizer capture ubiquitous statistical patterns in natural text, which is crucial for transformer language models to generalize unseen data.

# 7 Conclusion

This work set out to explain *how* and *why* larger vocabularies boost language model performance. Our experiments reveal a single, robust mechanism: enlarging the vocabulary reduces tokenized text complexity, making non-i.i.d. patterns easier to learn, better approximating natural data's low intrinsic complexity, and ultimately lowering language modeling difficulty. Once a vocabulary reaches roughly 24K size, every common word is already a single token; subsequent growth therefore does not refine segmentation but instead steepens the long-tailed frequency distribution, focusing optimization on the same frequent tokens and driving loss down. Norm-constraining that erases the frequency signal removes these gains, confirming causality, while enlarging model parameters with a fixed tokenizer reproduces the benefit, pointing to a shared optimization dynamic between vocabulary and model scaling. Recognising this fact turns Kolmogorov complexity of data into a principled dial for tokenizer–model co-design and sharpens our understanding of the scaling forces that govern language-model pre-training. We therefore conclude:

> *Expanding the tokenizer mainly reduces uncertainty for the most common words,*
> *with little payoff for the rare tail.*

## Acknowledgement

This work was supported by Institute for Information & communications Technology Planning & Evaluation (IITP) grant funded by the Korea government(MSIT) (RS-2019-II190075, Artificial Intelligence Graduate School Program(KAIST)), and Institute of Information & communications Technology Planning & Evaluation (IITP) grant funded by the Korea government(MSIT) (No.RS-2022-II220184, 2022-0-00184, Development and Study of AI Technologies to Inexpensively Conform to Evolving Policy on Ethics).

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

# A   Influence of token-frequency imbalance on unigram and language models

In this section, we provide a detailed explanation of our research question. Natural language exhibits strong contextual dependencies rather than behaving as an i.i.d. process: each token's probability depends on its preceding context. As a result, the entropy rate, $H_\infty = \lim_{t \to \infty} H(X_t \mid X_{<t})$, which captures the optimal per-token uncertainty in text, is strictly lower than the unigram Shannon entropy $H_1 = -\sum_w p(w) \log p(w)$. Although entropy rate and Shannon entropy coincide for truly i.i.d. data, in natural language, they can differ by several bits per token.

Under a pure unigram model, minimum cross-entropy loss exactly equals Shannon entropy, and modifying the vocabulary size of the tokenizer immediately changes the Shannon entropy. Typically, enlarging the vocabulary segments frequent multi-token patterns into single tokens, driving their individual relative frequency up and reducing Shannon entropy. But if the vocabulary size grows too large, the new entries tend to be rare tokens, which lengthen the tail and can actually increase the Shannon entropy as token-frequency imbalance rises [67]. However, experimental results show that expanding a BPE vocabulary to around $80K$ lowers the Shannon entropy of unigram models, demonstrating that a more skewed token-frequency distribution is advantageous at practical vocabulary scales [47]. This pattern extends to n-gram models as well: their conditional Shannon entropy—the lowest possible n-gram cross-entropy—can never exceed the unigram entropy, so lowering the unigram entropy necessarily lowers the conditional entropy.

Language model loss $\mathcal{L}(\theta)$ minimizes

$$\mathcal{L}(\theta) = H_\infty + \sum_{x \in V} p(x) \, D_{\mathrm{KL}}\big(P(\cdot \mid x_{<t}) \,\|\, Q_\theta(\cdot \mid x_{<t})\big). \tag{1}$$

where $V$ denotes the tokenizer's vocabulary, $p(x)$ the marginal probability of token $x$, $P(\cdot \mid x_{<t})$ the true next-token distribution given the full history $x_{<t}$, and $Q_\theta(\cdot \mid x_{<t})$ the model's predicted distribution with parameters $\theta$. When the target label is a one-hot vector, the language model loss can be written as $\mathcal{L}(\theta) = \sum_{x \in V} p(x) \big[ -\log Q_\theta(x \mid x_{<t}) \big]$ so it is a marginal-frequency-weighted average of the model's surprisal $-\log Q_\theta$. By contrast, the Shannon entropy of the unigram model is a marginal-frequency-weighted average of the self-information $-\log p(x)$ in the dataset. Even though frequent token logits and embedding norms are higher than rare ones (see Appendix D and E), the loss $-\log Q_\theta$ depends not only on the underlying relative token-frequencies but also on training dynamics as well. We therefore cannot derive a closed-form expression to predict how much the loss on frequent tokens shrinks versus how much the rare token losses grow under $\mathcal{L} = p_{\mathrm{freq}} L_{\mathrm{freq}} + p_{\mathrm{rare}} L_{\mathrm{rare}}$. This masking effect makes it substantially harder to measure the influence of token-frequency imbalance in language models than in the unigram and n-gram models.

# B  Coverage of the most frequent N words

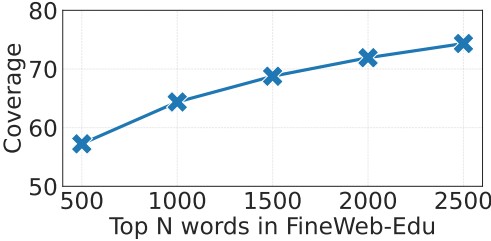
(a) Coverage of frequent words in FineWeb-Edu

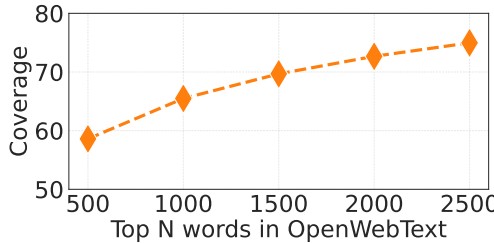
(b) Coverage of frequent words in OpenWebText

Figure 6: Figures 6a and 6b illustrate the cumulative coverage of the 2,500 most frequent words in the Fineweb-edu and OpenWebText datasets, respectively.

In this section, we measure the coverage of the frequent words in Fineweb-edu [43] and OpenWebText [15]. Both dataset exhibit a steep rise in cumulative coverage as we include more high-frequency tokens, but with subtly different baselines and slopes. In figure 6a, the most frequent 500 words already cover about 58% of all tokens, climbing to roughly 75% once we take the 2,500 most frequent words. OpenWebText (Figure 6b) starts marginally higher—around 59% at most frequent 500 words, but follows an almost identical trajectory, reaching about 76% coverage by the frequent 2,500 words. This pattern underscores how a relatively small core vocabulary captures the vast majority of running text in both corpora, with only modest gains as we move deeper into the long tail.

# C  OpenWebText experiments results

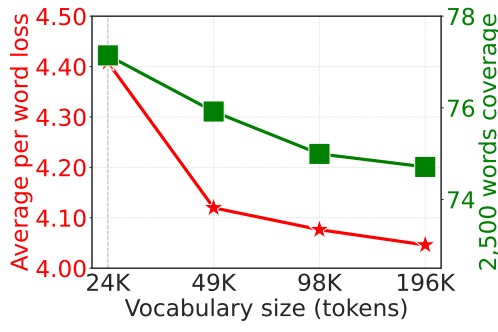
(a) Frequent word loss on an OpenWebText

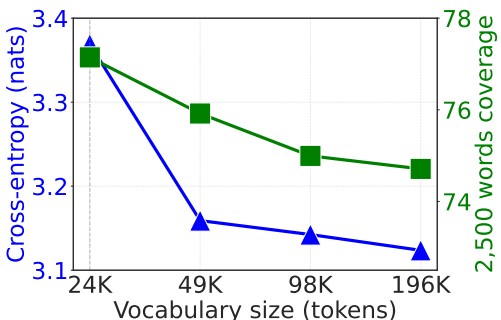
(b) Global cross-entropy on an OpenWebText

Figure 7: Figure 7a reveals that expanding the vocabulary from 24K to 196K steadily reduces the average per-vocabulary loss of high frequency words. Figure 7b indicates that the most frequent 2,500 tokens still contribute roughly 75 of the total loss, while the rare words losses grow with vocabulary size, similar to 2b. Figure 7b further shows that the global cross-entropy loss falls by about 0.25 nats as the vocabulary grows, demonstrating that the reduction of loss on frequent words outweighs the inflation of rare-token losses.

To verify that reducing frequent-word loss is not a by-product of dataset quality, we repeat the same experiments in section 3.6 on the OpenWebText dataset. Figure 7a shows that widening the vocabulary from 24K to 196K in OpenWebText progressively reduces the average loss assigned to high-frequency words. Figure 7b indicates that the most frequent 2,500 tokens still account for about 75 of the total loss, whereas the loss on infrequent tokens grows with vocabulary size, paralleling the pattern seen in Figure 2b. Figure 7b further demonstrates that the global cross-entropy loss falls from 3.37 nats at a 24K vocabulary to 3.12 nats at 196K, implying that the reduction in loss on frequent words more than offsets the increase in rare-token loss, regardless of dataset quality or type.

# D Constraining embedding norms increases cross-entropy

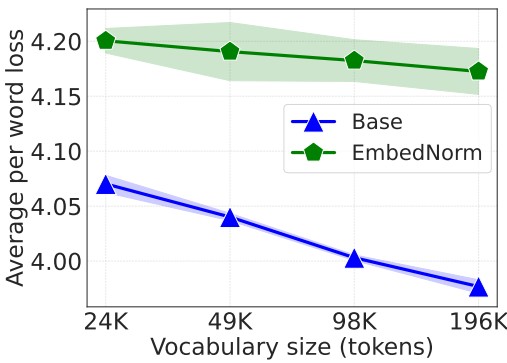
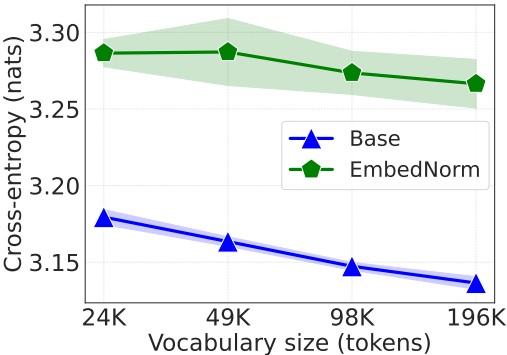

(a) Frequent word loss gap of base and embednorm    (b) Cross-entropy gap of base and embednorm

Figure 8: Constraining the norms of both input and output token embeddings to mitigate the impact of token-frequency imbalance during training increases average per-word loss on $2,500$ frequent words in the FineWeb-Edu as illustrated in figure 8a, which in turn drives up the global cross-entropy loss on the FineWeb-Edu (figure 8b).

Higher token-frequency imbalance provides plentiful training examples for frequent words but far fewer for rare ones. Section 3.6 demonstrates that this reduces loss on frequent words while inflating it on rare words. We evaluate whether reducing the impact of token-frequency imbalance during pre-training leads to a performance degradation.

In language models, the token embedding norm of input and output embeddings reflects the token-frequency imbalance in training data. For input embeddings, every time a token shows up in a training batch, its token embedding receives gradient updates, and tokens that do not appear only undergo weight decay, causing their embedding norms to shrink over time [30, 10]. In the output embedding layer, the target token in the output embeddings receives a stronger gradient than non target token embedding during training [39] [4] (see the Appendix E for a detail explanation). Thus embedding norm of a frequent token is typically larger in both input and output embedding. Even though language models with pre-LN [62] normalize the final layer hidden states, frequent tokens still develop larger output-embedding norms and highly align with the final layer hidden states, inflating their logits and predicted probabilities and ultimately reducing their average per-word loss.

To reduce the impact of token-frequency imbalance during pre-training, we constrain the token embeddings of the input and output embedding layer to a unit norm, since the token embeddings of our base model already average unit length. Figure 8a demonstrates that constraining the norms of both input and output token embeddings to address token-frequency imbalance during training leads to a higher average loss for $2,500$ frequent words. This increases the global cross-entropy on the training dataset and lowers downstream performance as illustrated in figure 8b and figure 4c respectively. This observation demonstrates that placing equal emphasis on rare and frequent tokens undermines performance; ensuring minimal uncertainty when predicting frequent tokens is essential.

---

[4] we assume input and output embeddings are untied from each other

# E    Frequent and rare token norm in output embedding

In this section, we explain why frequent tokens acquire larger output-embedding norms and logits by deriving and analysing the gradients of the output embedding. Cross-entropy loss with a vocabulary size $|V|$ and hidden dimension size $d_{\text{model}}$, using a one-hot target $t \in \mathbb{R}^{|V|}$, the logit vector is $\ell = hE_{\text{out}}^{\top}$ where $h \in \mathbb{R}^{d_{\text{model}}}$ is the final hidden state and $E_{\text{out}} \in \mathbb{R}^{|V| \times d_{\text{model}}} = [u_1, \ldots, u_{|V|}]$ is the output-embedding matrix. When input and output embeddings are untied, the gradient with respect to each row of $E_{\text{out}}$ decomposes into

$$\frac{\partial \mathcal{L}}{\partial E_{\text{out}_t}} = (p_t - 1)\, h, \qquad \frac{\partial \mathcal{L}}{\partial E_{\text{out}_j}} = p_j\, h \ \ (j \neq t), \tag{2}$$

where $p_t = \text{softmax}(\ell)_t$ [5, 39]. Because $p_t \ll 1$ at the start of training, $\left\| \partial \mathcal{L}/\partial E_{\text{out}_t} \right\|_2 \approx \|h\|_2$ while each non-target row scales only with $p_j \|h\|_2$ $(p_j < 1/|V|)$. Thus, every time token $t$ appears, its embedding is pulled almost $\|h\|_2$ units along $+h$, whereas each competing row is nudged by a factor of $p_j \ll 1$. As tokens recur in the training data, their token embeddings accumulate gradient updates roughly proportional to their counts, so $\|E_{\text{out}_t}\|_2$ grows in line with token-frequency. Since each logit factorizes as $\ell_t = \|h\|_2 \|E_{\text{out}_t}\|_2 \cos \theta_t$ and frequent tokens not only acquire the largest norms but also align closely with final hidden-state directions $\cos \theta_t \approx 1$, they end up with disproportionately large logits whereas rare tokens suffer both smaller norms and larger angular deviations [5, 30]. Although $\ell_2$ weight decay can slow this norm inflation, they merely dampen the effect rather than remove the underlying frequency norm logit correlation. This frequency-proportional amplification explains the empirical observation that output-embedding norms and softmax logits scale with token-frequency in standard language models.

# F    OLMo-2 result

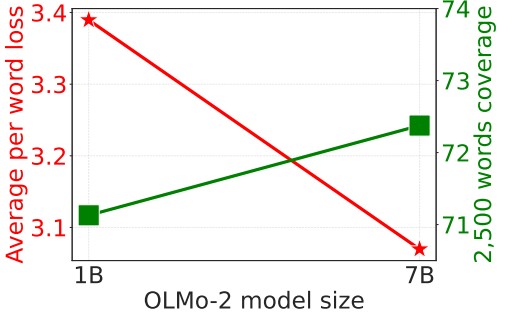
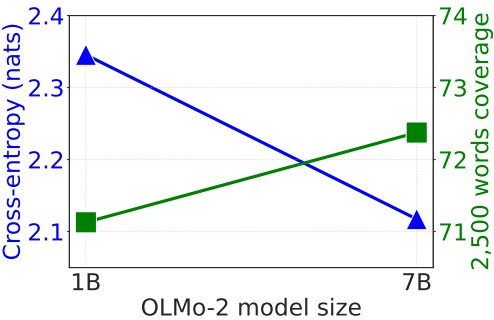

(a) Frequent word loss in OLMo-2 models    (b) Global cross-entropy in OLMo-2 models

Figure 9: Figure 9a illustrates that larger 7B model reduces average per-vocabulary loss from 3.39 nats to 3.07 nats while slightly increasing the proportion of loss covered by frequent words from 71% to 72.5%. Figure 9b further demonstrates that Scaling from 1B to 7B reduces the overall cross-entropy from 2.35 nats to 2.12 nats, confirming the same pattern persists in contemporary large-scale language models.

To identify whether the reduction in frequent words loss with increasing model size holds for contemporary large language models, we perform analogous experiments using the OLMo-2 series [41]. Figure 9a and 9b indicate that the average per vocabulary loss falls from 3.39 nats in the 1B parameter model to 3.07 nats in the 7B variant while slightly increasing the proportion of loss covered by 2,500 frequent words from 71% to 72.5%. Figure 9b further shows that global cross-entropy loss declines from 2.35 nats for OLMo-2 1B to 2.12 nats for OLMo-2 7B. Notably, OLMo-2 employs a much larger vocabulary (cl100K [42]) than Pythia (50304 tokens [7]) and trains on a larger corpus [41], which helps drive down the average loss on high-frequency words. These results confirm that the same trend holds for modern large-scale language models as well.

# G   Are the gains mainly driven by parameter increases from vocab expansion?

To probe whether gains come from parameter growth with larger vocabularies, we run a controlled study comparing a 24K vocabulary model (122M parameters) to a near equal size 49K model (124M parameters) to isolate embedding capacity effects. We reduced the hidden dimension from 768 to 648 and the feed-forward dimension from 2048 to 1728, while maintaining identical model depth which known to have a dominant impact on performance [59, 44]. As shown in the table 3, the 49K model incurs higher cross-entropy loss and delivers inferior results on downstream benchmarks (ARC-E, HellaSWAG, PIQA, and SCIQ) compared to its 24K counterpart, indicating that simply expanding the embedding layer contributes far less to model expressivity than is often assumed.

Table 3: A 24K vs. size-matched 49K indicates embedding expansion yields limited gains.

| Vocab (Params) | CE Loss | Downstream (%) |
|---|---|---|
| 24K (122M) | 3.179 | 54.73 |
| 49K (124M) | 3.563 | 50.68 |
| 49K (161M) | 3.171 | 55.44 |

# H   Benefits of larger vocabularies hold in tied embedding models?

Several studies show that tying input and output embeddings can change pre-training dynamics [24, 32]. We therefore repeat our study with tied embeddings and report average frequent-word loss and cross-entropy on FineWeb-Edu in the table 4. The tied embedding variant shows broadly similar average frequent word loss and ce-loss to the untied baseline.

Table 4: Average frequent-word loss and cross-entropy of tied and untied embedding models

|  |  | 24K | 49K | 98K | 196K |
|---|---|---|---|---|---|
| **Tied** | Avg frequent word loss | 4.067 | 4.029 | 3.991 | 3.968 |
|  | Cross entropy | 3.177 | 3.160 | 3.146 | 3.134 |
| **Untied** | Avg frequent word loss | 4.074 | 4.031 | 4.001 | 3.977 |
|  | Cross entropy | 3.179 | 3.163 | 3.147 | 3.136 |

# I  Learning rate exploration

We use a fixed learning rate of $6 \times 10^{-4}$ for an 85M non-embedding–parameter language model, mirroring the learning rate of GPT-3 with an equivalent non-embedding model size [8]. we further conducted a learning-rate exploration with four additional learning rates: $2.4 \times 10^{-3}$, $1.2 \times 10^{-3}$, $1.5 \times 10^{-4}$, $7.5 \times 10^{-5}$. Table 5 and 6 report the resulting cross-entropy loss and average word loss of frequent 2500 words in fineweb-edu across different vocabulary sizes and learning rates, using the same validation dataset described in Section 3.1.

Table 5: Cross-entropy loss across five learning rates.

| Cross-Entropy Loss | 2.4e-3 | 1.2e-3 | 6.0e-4 | 1.5e-4 | 7.5e-5 |
|---|---|---|---|---|---|
| 24K | 3.156 | 3.141 | 3.179 | 3.484 | 3.779 |
| 49K | 3.135 | 3.114 | 3.163 | 3.461 | 3.751 |
| 98K | 3.122 | 3.099 | 3.147 | 3.443 | 3.736 |
| 196K | 3.111 | 3.086 | 3.136 | 3.429 | 3.725 |

Table 6: Average frequent-word loss across five learning rates.

| Frequent Word Loss | 2.4e-3 | 1.2e-3 | 6.0e-4 | 1.5e-4 | 7.5e-5 |
|---|---|---|---|---|---|
| 24K | 4.033 | 4.012 | 4.074 | 4.431 | 4.783 |
| 49K | 3.990 | 3.970 | 4.031 | 4.399 | 4.749 |
| 98K | 3.963 | 3.939 | 4.001 | 4.373 | 4.725 |
| 196K | 3.942 | 3.916 | 3.977 | 4.352 | 4.703 |

# J  Detailed Experimental Setting

In this section, we provide detailed configurations of pretraining to reproduce our results. The training setup (Table 7) uses a global batch size of 256, weight decay 0.1, and sequence length 2048. Optimization is Adam with a cosine learning-rate schedule, a 700-step warmup, and a weight-initialization scale of 0.02. The model setup (Table 8) covers two different model size: an 85M model with 12 layers and 12 heads ($d_{\text{model}} = 768$, $d_{\text{ffn}} = 2048$, $d_{\text{head}} = 64$) and a 450M model with 21 layers and 21 heads ($d_{\text{model}} = 1344$, $d_{\text{ffn}} = 3548$, $d_{\text{head}} = 21$). Together, these tables specify the standardized training hyperparameters and the core architectural dimensions for both scales.

Table 7: Training configurations. LR Schedule denotes learning-rate schedule.

| Global Batch Size | Weight Decay | Sequence Length | Optimizer |
|---|---|---|---|
| 256 | 0.1 | 2048 | AdamW |

| LR Schedule | Warmup | Weight Init. |
|---|---|---|
| Cosine | 700 steps | 0.02 |

Table 8: Model configurations.

| Size | $n_{\text{layers}}$ | $n_{\text{heads}}$ | $d_{\text{model}}$ | $d_{\text{ffn}}$ | $d_{\text{head}}$ |
|---|---|---|---|---|---|
| 85M | 12 | 12 | 768 | 2048 | 64 |
| 450M | 21 | 21 | 1344 | 3548 | 21 |

## K    How much increasing the model size lower the frequent and rare word loss?

We conducted experiments to quantify scaling effects on frequent and rare words loss and demonstrate that scaling model capacity disproportionately benefits the prediction of high-frequency tokens relative to low-frequency tokens. Since BPE vocabulary IDs are not token frequency ordered, we cannot directly distinguish common and rare tokens in the Pythia and OLMo-2 training corpora. To proxy frequency, we utilize the Google Web Trillion Word Corpus frequency list, which records counts for $333,333$ unique words appearing at least $10,000$ times in a $1.0249 \times 10^{12}$ words. We classify the top $2,500$ entries as frequent words and the bottom $10,000$ as rare words. As shown in the table 9, although cross-entropy loss declines for both groups with increasing model size, the reduction for frequent tokens is markedly greater. We therefore conclude that the dominant effect of model scaling is the lowering of frequent-token loss, which in turn drives the overall cross-entropy reduction.

Table 9: Frequent, rare word loss and cross-entropy loss of Pythia and OLMo-2

| Model | Params | Frequent word loss | Rare word loss | CE-loss |
|---|---|---|---|---|
| Pythia | 1B | 3.01 | 4.99 | 2.51 |
| | 7B | 2.55 | 4.93 | 2.26 |
| OLMo-2 | 1B | 2.60 | 4.27 | 2.22 |
| | 7B | 2.20 | 4.19 | 2.00 |

