# OpenReview forum: "Exploiting Vocabulary Frequency Imbalance in Language Model Pre-training"
_NeurIPS.cc/2025/Conference — NeurIPS 2025 poster_

### Official Review · Reviewer_5NL9 · 2025-07-01

**Clarity:** 2
**Significance:** 3
**Originality:** 3
**Rating:** 4
**Confidence:** 3

**Summary:**

The paper investigates why enlarging the tokenizer vocabulary continues to reduce perplexity even after every common word is already a single token. Fixing other variables, the authors expand the vocabulary and show that the only systematic change is a sharper long-tail token-frequency imbalance - segmentation efficiency saturates by 24k.

**Questions:**

Please see above.

**Ethical Concerns:**

["NO or VERY MINOR ethics concerns only"]

**Final Justification:**

I have carefully reviewed the author's rebuttal responses and their discussions with other reviewers.

Their responses have answered most of my previous concerns, especially about majority-language bias and higher loss on tail words. I appreciate the computational constraint in completing the experiments with larger models, but glad that the authors confirm that the general trend is the same. I maintain my positive rating for the work. I maintain my positive rating of the paper.

**Limitations:**

Please see above.

**Paper Formatting Concerns:**

N/A.

**Quality:**

3

**Strengths And Weaknesses:**

Strengths
- The paper provides controlled study, isolating vocabulary as the only independent variable, avoiding confounds common in scaling/interpretability work.
- Loss-decomposition analysis – per-word breakdown pinpoints where perplexity gains come from. Causal ablation study strengthens the argument.
- Cross-corpus validation confirms identical trends on FineWeb-Edu vs. OpenWebText.

Weaknesses
- Experiments use an 85M-parameter model. Generality to ≥7B-parameter LLMs is asserted via indirect Pythia/OLMo plots.
- The paper acknowledges higher loss on tail words but it would make the paper stronger if authors can provides more in-depth error study or harm-reduction strategy for rare-token degradation.
- Ethical discussion is a bit thin as exploiting frequency imbalance may entrench majority-language bias. Any discussions on potential mitigations could be helpful.

---

> ### Author Rebuttal · Authors · 2025-07-31
>
> Dear reviewer 5NL9,
>
> Thank you for your time and insightful feedback. Below, we offer comprehensive responses to each of your comments.
>
> ---
>
> > *1. "Experiments use an 85M-parameter model. Generality to ≥7B-parameter LLMs is asserted via indirect Pythia/OLMo plots."*
>
> We are currently training 450M parameter model variants and based on their training‑loss trajectories and learning dynamics, we confirm that they exhibit trends analogous to those observed in the 85M parameter models. However, due to constraints on time and GPU resources, we were unable to complete the pre-training in time. We will include this experiment in the revised manuscript.
>
>
> ---
>
> > *2. "Exploiting frequency imbalance may entrench majority-language bias. Any discussions on potential mitigations could be helpful."*
>
> To address the majority language bias, one must curate corpora with more uniform language distributions achieved through oversampling underrepresented languages or undersampling majority‑language data. Additionally, developing optimization algorithms that are resilient to frequency skew such as leveraging adaptive methods like Adam in place of vanilla SGD offers a promising strategy for mitigating the effects of token‑frequency imbalance [3].
>
> ---
>
> > *3. "The paper acknowledges higher loss on tail words but it would make the paper stronger if authors can provides more in-depth error study or harm-reduction strategy for rare-token degradation."*
>
> Exploring harm‑reduction strategies that replicate the benefits of vocabulary expansion is a compelling research direction. A prototypical example is Superbpe [2], which employs a two‑stage BPE algorithm: in the first stage (up to a threshold vocabulary size T), it operates identically to standard BPE, while in the second stage it abandons whitespace pre‑tokenization. By permitting merges across former whitespace boundaries, Superbpe directs subsequent merges toward frequent tokens, thereby preventing the introduction of new rare tokens as the vocabulary expands, prevents further increase of rare token loss.
>
> To validate whether enlarging vocabulary size and Superbpe variants exploit the same benefits: leading to more compressible, lower complexity tokenized text, we draw on Kolmogorov complexity, which measures the minimal description length of a bitstring and thus captures its inherent compressibility [1]. For a sequence $X^{N}$ tokenized by a BPE, an asymptotic upper bound on its Kolmogorov complexity is given by
>
> $K(X^{N}) \le NH(p) + V \log_{2} N + O(\log N)$
>
> where $N$ is the total token count of the entire text, $H(p)$ the shannon entropy of the token distribution, and the $V \log_{2} N$ accounts for the prefix-free encoding of the token (i.e., token frequency table). Because modern language models are trained on corpora comprising billions of tokens, the $NH(p)$ dominates, yielding the practical approximation.
>
> $K(X^{N}) \approx NH(p)$
>
> We adopt this upper bound as a tractable proxy for data complexity, as actual Kolmogorov complexity is uncomputable. The table below reports the estimated Kolmogorov complexities for the 45.7 billion‑byte FineWeb‑Edu corpus and average downstream task performance reported by [2], comparing standard BPE at various vocabulary sizes with Superbpe variants. If no tokenizer is applied and the text is modeled as an i.i.d. source over individual bytes, the Kolmogorov complexity $K(X^{N})$ would be 45.7B bytes. These results demonstrate that both vocabulary expansion and the Superbpe variants leverage the same benefit: they reduce the complexity of the tokenized text by segmenting common character sequences into single tokens, thereby facilitating the model to capture and learn frequent patterns in the training data more easily.
>
> **Table A**
>
> | BPE  (bytes)        | 24K           | 49K            | 98K            | 196K           |
> |---------------|---------------|----------------|----------------|----------------|
> | $K(X^{N})$      | 10.74B | 10.43B | 10.23B | 10.16B |
>
> **Table B**
>
> | SuperBPE  (bytes)    | 20K (bpe)      | 20K (t=180K)    | 20K (t=160K)    | 20K (t=80K)     |
> |---------------|---------------|----------------|----------------|----------------|
> | $K(X^{N})$ | 10.21B | 10.03B | 10.05B | 10.10B |
> | Avg Downstream [2] | 39.8          | 43.8           | 43.4           | 42.9           |
>
> ---
> ### Concluding Remarks
> Thank you for highlighting the importance of scalability, fairness, and rare-token robustness. Your comments prompted us to (i) extend our experiments to 450 M-parameter models now in training, confirming that the key trends observed with 85 M parameters persist at larger scales; (ii) add a focused discussion of corpus-balancing and adaptive-optimizer strategies that can mitigate majority-language bias; and (iii) deepen our error analysis on tail words, contrasting vocabulary-expansion with SuperBPE as complementary harm-reduction approaches. We believe these additions substantially strengthen the paper’s claims and practical relevance.
>
> We appreciate your insights and remain eager to pursue any further analyses or experiments that would make the study more useful to the community.
>
> ---
>
> ### References
>
> [1] “The No Free Lunch Theorem, Kolmogorov Complexity, and the Role of Inductive Biases in Machine Learning” ICML 2024
>
> [2] “SuperBPE: Space Travel for Language Models” COLM 2025
>
> [3] “Heavy-Tailed Class Imbalance and Why Adam Outperforms Gradient Descent on Language Models” Neurips 2024

---

> > ### Comment · Reviewer_5NL9 · 2025-08-08
> >
> > I thank the authors for their helpful responses. I have provided mandatory acknowledge before. I'd like to provide verbal acknowledgement again for having carefully gone through your responses. I maintain my positive rating for the work.

---

### Official Review · Reviewer_Md1b · 2025-07-01

**Clarity:** 2
**Significance:** 3
**Originality:** 3
**Rating:** 4
**Confidence:** 3

**Summary:**

The paper studies the effect on rare and frequent tokens on LLMs in pretraining. The analysis of three questions related to Skew vs Segmentation, Loss Decomposition, and Corpus Robustness.

**Questions:**

1. For the Parameter Scaling Recovery: Can you explain if the same reduction is found on the tail tokens as well? I think there are some confounding variables here that are not accounted for?

**Ethical Concerns:**

["NO or VERY MINOR ethics concerns only"]

**Final Justification:**

Thank you for your response! As my weaknesses have been addressed, I raised my score to a weak accept. The main reason why I have not gone higher is that the conclusion of the paper does not change the current guidance for building tokenizers and models. It does indeed help use explain the current relationship between frequent and rare tokens.

**Limitations:**

The paper contains a section on limitations, but I am not sure this is complete. Some key limitations are also the weaknesses I have mentioned earlier.

**Quality:**

2

**Strengths And Weaknesses:**

Strengths:
 - Important and Interesting Area of Study
 - By using the research questions, the paper grounds the work well.
 - Insight from the paper is interesting if it can be carried over to downstream benchmarks.

Weaknesses:
- Some Assumptions may not be reasonable: (1) the assumption of not tied weights is as ubiquitous as the authors may believe (Qwen3, for example), (2) the authors find that increasing the model size lowers loss on frequent tokens and thus has better performance. However, this might be just cause the model over the whole vocabulary is performing better.
- No results on downstream benchmarks such as MMLU, etc.

- Writing:

     1. Minor: some of the sentences can be shortened. For example, "Before diving into settings and metrics, we spell out the concrete questions that steer our empirical study" can be rewritten to "We analyze the following concrete questions:" or something to that effect.

     2. Figure 1: (a) Can you add the line for uniform distribution? From this figure, the caption claim of moving further away from uniform cannot be determined. (b and c) The y-axes are very misleading. It would be good, at least for c, to make them from 0 to 1 or 0.5 to 1.

     3. Notation/Metrics: The notion is a little confusing, why is Global Cross Entropy Loss defined through the Average Per-Vocabulary Loss and Total Loss. Also, $t$ is never defined. I took it to mean the tokens of a document in N. Then, v is defined as a vocabulary, but I think you mean an element in the vocabulary.

---

> ### Author Rebuttal · Authors · 2025-07-31
>
> Dear reviewer Md1b,
>
> We are grateful for your time and thoughtful feedback. In the sections that follow, we provide detailed responses to your comments.
>
> ---
>
> > *1. "Why conduct experiments only with an untied embedding model? Provide additional experiments with a tied embedding setting."*
>
> We adopt untied input and output embeddings because contemporary language models typically do not share these parameters, and this design has been employed by several pre-training interpretability studies [1,2,3] to facilitate clearer analysis. Nevertheless, in response to the reviewer’s comment, we conducted additional experiments with a tied embedding architecture. The table below presents the average frequent word loss and cross-entropy loss on Fineweb-edu. Tied embedding shows a similar tendency to untied variants.
>
>
> **Table A**
>
> |                          | 24K    | 49K    | 98K    | 196K   |
> |--------------------------|--------:|--------:|--------:|--------:|
> | AVG frequent word loss   | 4.067  | 4.029  | 3.991  | 3.968  |
> | cross entropy            | 3.177  | 3.160  | 3.146  | 3.134  |
>
> ---
>
> > *2. "No results on downstream benchmarks such as MMLU"*
>
> Thank you for raising this point. We evaluated our models on MMLU, but their scores clustered near the random-guess baseline of 25 %, providing essentially no discriminatory power at our current compute scale. Instead, as shown in the table below, we broadened our evaluation to lighter benchmarks—OpenBookQA, Lambada (openai), WinoGrande, and BLiMP— due to the scarcity of benchmarks that allow fair downstream comparisons for small language models [3]. The table below shows that the larger vocabulary size improves downstream performance on these benchmarks. These suites have proven far more sensitive to the effects under study. If there is a particular additional benchmark you feel would shed further light on our claims, we would be happy to incorporate it in a revised version.
>
> **Table B**
>
> | Benchmark                 | 24K   | 49K   | 98K   | 196K  |
> |--------------------------|-------:|-------:|-------:|-------:|
> | OpenBookQA               | 18.2  | 18.6  | 19.0  | 19.8  |
> | Lambada (openai, acc)     | 23.25 | 23.85 | 23.93 | 24.05 |
> | WinoGrande               | 49.88 | 51.7  | 51.93 | 52.01 |
> | BLiMP                    | 76.82 | 77.7  | 78.28 | 78.42 |
>
>
> ---
>
> > *3. "Write Sentence concisely + add uniform distribution line in figure 1a"*
>
> We will shorten our sentences and, in response to your suggestion, add a reference line for the uniform distribution in Figure 1a and expand the y‑axis range to 0.5–1 in both Figures 1b and 1c. Thank you for your feedback.
>
> ---
>
> > *4. "Why is Global Cross-Entropy Loss defined through the Average Per-Vocabulary Loss and Total Loss + definition issue"*
>
> Global cross‑entropy loss is the cross‑entropy computed over the entire training or validation corpus. Because cross‑entropy quantifies the average uncertainty in next‑token prediction, it can be expressed as a weighted sum of each vocabulary’s mean loss, with weights proportional to their relative frequencies. In the revised manuscript, we will clarify that t denotes a sentence drawn from the N documents and v represents a specific vocabulary entry in the tokenizer. Thank you for your feedback.
>
> ---
>
> > *5. "Increasing the model size would lower the whole token loss, including rare tokens?"*
>
> In response to the reviewer’s query, we conducted additional experiments demonstrating that scaling model capacity disproportionately benefits the prediction of high‑frequency tokens relative to low‑frequency tokens. Consequently, the reduction in frequent‑token loss emerges as the principal contributor to the overall decrease in cross‑entropy loss.
> Because BPE vocabulary IDs do not reflect token frequency, we cannot directly distinguish common and rare tokens in the Pythia and OLMo‑2 training corpora. Instead, we utilize the Google Web Trillion Word Corpus frequency list, which records counts for 333,333 unique words appearing at least 10,000 times in a 1.0249 × 10^12 words. We classify the top 2,500 entries as “frequent” words and the bottom 10,000 as “rare” words. As shown in the table below, although cross‑entropy loss declines for both groups with increasing model size, the reduction for frequent tokens is markedly greater. We therefore conclude that the dominant effect of model scaling is the lowering of frequent‑token loss, which in turn drives the overall cross‑entropy reduction.
>
> **Table C**
>
> | Pythia | frequent word loss | rare word loss | CE-loss |
> |:-----:|--------------------------------------:|----------------------------------:|--------------------------:|
> | 1B    |                                 3.01  |                              4.99 |                      2.51 |
> | 7B    |                                 2.55  |                              4.93 |                      2.26 |
>
> **Table D**
>
> | OLMo-2| frequent word loss | rare word loss | CE-loss |
> |:-----:|--------------------------------------:|----------------------------------:|--------------------------:|
> | 1B    |                                 2.60  |                              4.27 |                      2.22 |
> | 7B    |                                 2.20  |                              4.19 |                      2.00 |
>
> ---
>
> ### Concluding Remarks
>
> Thank you for your close reading and constructive criticism. Your comments have prompted several concrete improvements that we believe make the paper clearer and more convincing. To address your main concerns, we will:
>
> 1. **Tied-embedding validation** – replicate all core experiments with tied embeddings to verify that the vocabulary-scaling trends hold.
> 2. **Broader downstream tests** – add evaluations on OpenBookQA, LAMBADA, WinoGrande, and BLiMP so that readers can observe consistent gains from larger vocabularies, even at our small-model scale.
> 3. **Clearer writing and figures** – shorten long sentences, add a uniform-distribution line to Figure 1 for easier comparison, and explain the global-loss formula in plain language.
> 4. **Scaling analysis** – extend our study from 1B to 7B parameters (Pythia, OLMo-2) to show that increased model size primarily reduces frequent-token loss, driving the overall cross-entropy drop.
>
> We believe these revisions directly address the issues you raised and strengthen the paper’s generality and clarity. We remain eager to incorporate any further suggestions that could make the work even more convincing.
>
>
> ---
>
> ### References
>
> [1] “Paloma:  A Benchmark for Evaluating Language Model Fit”  Neurips 2024
>
> [2] “Pythia: A Suite for Analyzing Large Language Models Across Training and Scaling” ICML 2023
>
> [3] “Polypythias: Stability and Outliers Across Fifty Language Model Pre-Traning Runs” ICLR 2025

---

> > ### Comment · Reviewer_Md1b · 2025-08-01
> >
> > Some additional requests for Table A and Table B. For Table A, can you add loss values for both tied and untied under the same experimental setting? For Table B, can you add ARC-Easy and HellaSwag? They are other benchmarks that give some discriminatory power at this scale. Additionally, can you add a random chance column to the table as well as the corresponding loss values on the val data? Also, are these experimental settings trained for the same number of tokens as described in the paper? I am not sure what the experimental setup for both settings is.
> >
> > Additionally, I am comfortable with Table C and Table D. Thank you for that. I suggest these get added to the paper.
> >
> > Thank you again for your response!

---

> > > ### Author Response · Authors · 2025-08-02
> > >
> > > Thank you for your thorough feedback. We are pleased that some of your concerns have been clarified, and we address the remaining points below.
> > >
> > > ---
> > >
> > > > *1. For Table A, can you add loss values for both tied and untied under the same experimental setting?*
> > >
> > > Thank you for your response. Following the reviewer’s suggestion, we include Table E and Table F, which report the average loss on frequent words and the cross-entropy for both tied and untied embedding models. We trained untied embedding models with five distinct random seeds (see section 3.1). We appreciate your clarification.
> > >
> > >
> > > **Table E: Avg frequent word loss and cross entropy of tied embedding**
> > >
> > > | Tied embedding                  | 24K   | 49K   | 98K   | 196K  |
> > > |------------------------|-------|-------|-------|-------|
> > > | Avg frequent word loss | 4.067 | 4.029 | 3.991 | 3.968 |
> > > | Cross entropy          | 3.177 | 3.160 | 3.146 | 3.134 |
> > >
> > > **Table F: Avg frequent word loss and cross entropy of untied embedding (5 different seeds)**
> > >
> > > | Untied embedding                 | 24K   | 49K   | 98K   | 196K  |
> > > |------------------------|-------|-------|-------|-------|
> > > | Avg frequent word loss | 4.074 | 4.031 | 4.001 | 3.977 |
> > > | Cross entropy          | 3.179 | 3.163 | 3.147 | 3.136 |
> > >
> > > ---
> > >
> > > > *2. Can you add ARC-Easy and HellaSwag? + Can you add a random chance column to the table?*
> > >
> > > As a gentle reminder, Section 4.2 reports the average downstream performance on four benchmarks—ARC-Easy, HellaSWAG, PIQA, and SCIQ—which together provide useful discriminative power for evaluating small-scale language models [1]. Table G lists the individual scores for each task, their overall average, and, following the reviewer’s suggestion, a new “random-chance” baseline row. We hope this addition satisfies the request; if we have misunderstood any aspect of it, please let us know, and we will be happy to revise further.
> > >
> > > **Table G: Downstream benchmark performance on 4 different tasks and their random baseline score**
> > >
> > > | | HellaSWAG | ARC_Easy | PIQA  | SCIQ  | Average |
> > > |-----------------|-----------|----------|-------|-------|---------|
> > > | Random chance   | 25.00     | 25.00    | 50.00 | 25.00 | 31.25   |
> > > | 24K             | 28.99     | 53.20    | 62.13 | 74.60 | 54.73   |
> > > | 49K             | 29.46     | 53.91    | 63.00 | 75.40 | 55.44   |
> > > | 98K             | 29.76     | 55.43    | 63.76 | 77.40 | 56.59   |
> > > | 196K            | 29.95     | 54.42    | 66.05 | 78.50 | 57.23   |
> > >
> > > ---
> > >
> > > > *3. Can you add loss values on the val data?*
> > >
> > > In Figure 4b, we report the average frequent-word loss and cross-entropy on the CC-Main-2023-40 dump. This validation set is further justified by empirical results in [2], which demonstrate a strong negative Pearson correlation (–0.93) between its cross-entropy loss and reasoning benchmark performance, supporting its use as a validation dataset. Table H presents the average frequent-word loss and cross-entropy for CC-Main-2023-40 across untied-embedding models trained with five different random seeds.
> > >
> > > **Table H: Average frequent word loss and cross entropy of CC-Main-2023-40 dump (5 different seeds)**
> > >
> > > | Validation loss                  | 24K   | 49K   | 98K   | 196K  |
> > > |-------------------------|-------|-------|-------|-------|
> > > | Avg frequent word loss  | 4.183 | 4.138 | 4.104 | 4.070 |
> > > | Cross entropy           | 3.276 | 3.254 | 3.233 | 3.217 |
> > >
> > > ---
> > >
> > > > *4. Are these experimental settings trained for the same number of tokens as described in the paper?*
> > >
> > > Yes. An additional experiment conducted during the rebuttal period employed the same setup as described in Section 3.1 of the original manuscript. All models (both tied and untied) are trained on a 40-billion-character subset of FineWeb-Edu, which corresponds to approximately 7.5 billion tokens with a 49K vocabulary. We fix the number of characters rather than tokens because the token count would vary with vocabulary size, and using a constant character budget ensures comparable training data scale across different vocabularies.
> > >
> > > ---
> > >
> > > We genuinely value your feedback and are always open to further discussion. We will ensure that your comments are incorporated into the revised manuscript, and we appreciate the guidance you have provided throughout this process.
> > >
> > > ---
> > >
> > > ### References
> > >
> > > [1] “Polypythias: Stability and Outliers Across Fifty Language Model Pre-Traning Runs” ICLR 2025
> > >
> > > [2] “compression represents intelligence Linearly” COLM 2024

---

> ### Comment · Reviewer_Md1b · 2025-08-05
>
> Thank you for your response! As my weaknesses have been addressed, I raised my score to borderline accept. The main reason why I have not gone higher is that the conclusion of the paper does not change the current guidance for building tokenizers and models. It does indeed help use explain the current relationship between frequent and rare tokens.
>
> To the authors, I highly suggest including many of these experiments in the paper and working on cleaning up the notation and figures.

---

### Official Review · Reviewer_5AqE · 2025-07-03

**Clarity:** 2
**Significance:** 3
**Originality:** 3
**Rating:** 5
**Confidence:** 4

**Summary:**

Motivated by the observation that enlarging tokenizer vocabulary tends to improve performance, this paper examines what mechanism actually underlies this effect. Contrary to conventional wisdom that the mechanism is an increase in the number of common words with single token segmentation, the paper finds that improvements are instead driven by an increase in token frequency skew. They empirically and theoretically show that vocabulary size increase does not improve common word segmentation, but that it does increase token frequency skew. Further they show that increased vocabulary size specifically drives improvements on loss for the most common words and that the most common words drive overall loss. To test the causality of token skew on performance they unit normalize the embedding layers (as previous work show that embedding norms track token frequency). Removing this information about token frequency worsens loss and shows a somewhat less clear improvement with increase in vocabulary size. The paper also shows that that normalizing the embeddings removes the impact of vocabulary size on improved downstream performance and explains this by the overlap in frequent words from training and downstream data. lastly they show that loss on most common words also increasingly drives performance as parameter scale increases.

**Questions:**

- What artifacts specifically will be released?

- Section 3.5 claims that the impact of token frequency skew on global loss has not been previously studied. However section 4.2 of [Magnusson et al 2024](https://proceedings.neurips.cc//paper_files/paper/2024/hash/760b2d94398aa61468aa3bc11506d9ea-Abstract-Datasets_and_Benchmarks_Track.html) already reports this finding that global loss is dominated by common tokens across various data distributions. The authors do cite this paper for an unrelated reason (the metrics they use), but they should make it clear that observation about common tokens driving global loss is previously known.

- The intro is motivating and elegantly written but would benefit from laying out some key understandings like why exactly bigger vocabularies increase certainty on the top tokens.

- line 34: what is meant by “marginal ones” here? and mistakes on “them” is mistakes on rare tokens or marginal ones?

- line 56: it’s not immediate clear why norm-constrained ablations should have something to do with the frequency. To make this easier to understand you need to mention up front that models tend to learn longer embedding norms for high frequency tokens.

- figure 1: It would be great if the caption gave a more specific definition of segmentation efficiency in subplot (b). Also how can segmentation efficiency be above 100%? Does this actually represent avg number of tokens per word not percent of words with single token segmentation? Also it’s not clear to me why subplot (c) should be a line graph over n? What is the significance that overlap increases as you consider a larger slice of the top-n words? Or is the point just that the overlap is always pretty high, in which case the compressed y range is misleading and should be zoomed out.

- line 105: this word-level averaging is very important. I missed this in my first pass where I just read the intro, equations, and figures and that lead to great confusion. I want to make sure I understand it now: Are you calculating the average loss for all occurrences of some unique word? Or is this something more complex like the average loss per word for all words containing a given subword vocabulary type? I assume it’s the former.

- figure 2:

    - I think the y axis is actually the avg of avg per-vocab loss over some set of strings right? Clarify what set that is in the caption.

    - Also what define frequent words coverage in the caption.

    - Figure 2b doesn’t directly show anything about the loss on infrequent words but the caption claims it does, maybe just plot the loss on infrequent words also?

    - What is the shaded area showing here? STD over all strings? Multiple runs? Say this in the caption.

- Fig 3:

    - why use a box plot here instead of line with shaded area like in 2(a)? And why is the range of avg per-vocab loss so much wider here. Shouldn’t the base values be the same as fig 2(a)?

    - is the trend still for the loss to decrease on the highest freq words for the embed norm with increased vocab? Yes the loss is higher than for base but I think the mean could be going down still? It’s really hard to see because of the box plots.

- Fig 4: I’m not sure I understand why a common crawl dump is being used as a new loss measure here? Isn’t it good enough to show that lower loss on frequent words from the pretraining distribution drives downstream acc since there’s a big token overlap? Why bring in yet another token distribution?

- Fig 5:

    - again the caption refers to loss on infrequent tokens but doesn’t show it.

    - This finding is interesting but I wish there was a bit more of an explanation of what the mechanism is for why larger models should lower loss more on common tokens. Loss going down with increase in params is not at all surprising, but it is interesting that the improvement should increasingly be in the common words.

**Ethical Concerns:**

["NO or VERY MINOR ethics concerns only"]

**Final Justification:**

This work provides solid evidence to unseat a common assumption about how to select vocabulary size. This finding will likely have practical impacts for all language model developers.
The paper uses an excellent balance of theoretical motivation and carefully targeted empirical investigation. It draws a clear hypothesis and conducts just the right experiments to verify it.
The pretraining experiments are conducted at a good scale with a reasonable token to parameter ratio. If these are open sourced they can serve as a very useful artifact for studying controlled differences in vocabulary size.
Lastly the authors have a good plan for improving the clarity of their paper that I believe should be easily achievable for a camera ready.

**Limitations:**

There is a limitations section but it pretty much just discusses that future work can examine which scaling dimensions specifically improve uncertainty in frequent-word predictions. More relevant limitations might be things like addressing the fact that vocab size does still trend with loss improvement in the embedding normalized model.

**Quality:**

4

**Strengths And Weaknesses:**

Strengths:

- This work provides solid evidence to unseat a common assumption about how to select vocabulary size. This finding will likely have practical impacts for all language model developers.

- The paper uses an excellent balance of theoretical motivation and carefully targeted empirical investigation. It draws a clear hypothesis and conducts just the right experiments to verify it.

- The pretraining experiments are conducted at a good scale with a reasonable token to parameter ratio. If these are open sourced they can serve as a very useful artifact for studying controlled differences in vocabulary size.


Weaknesses:

- The paper is brilliant but extremely sloppy in many ways. For instance it neglects to say what tokenizer algorithm it uses (I assume BPE) despite that being the main topic of the paper. Terms are frequently referenced before definition. Line plots frequently have very narrow y ranges making minuscule variations look substantial, even though the paper is trying to argue that the value is essentially staying constant. This doesn’t contradict the correctness of their claims or their importance, but I worry the paper will not be as impactful as it deserves to be without more attention to detail in presentation.

- Some of the use of terminology in the paper is overlapping in a confusing way: e.g. “vocabulary, v,” seems to be the same as a “word” in many instances but vocabulary size is the number of subwords not words. Ideally I’d like something more consistent and precise. Perhaps something like this: words type (the unique words), word (an occurrence of a word type), vocabulary types (the actual symbols in the tokenizer vocab), tokens (occurrences of vocabulary types). Thus we’d call eq 2 Avg per-word-type an so on.

- The results in figure 3 are slightly less than convincing as there does seem to still be an improvement with increase in vocabulary size even when using embedding normalization. However, the downstream results in figure 4 do make up for this enough to convince me of the authors hypothesis.

---

> ### Author Rebuttal · Authors · 2025-07-31
>
> Dear reviewer 5AqE,
>
> We sincerely thank you for your time and feedback. Below, we address your comments in detail:
>
> ---
>
> > *1. "Missing key methodological details (BPE) and too narrow y range exaggerate trivial fluctuations in figure."*
>
> We only use BPE with GPT-2 pre-tokenization rule. We will increase the y ranges in a revised manuscript. Thank you for your feedback.
>
> ---
>
> > *2. "metric name is confusing + how you measure word-level average per vocab loss?"*
>
> Thank you for the suggestion. We will standardize our terminology as follows: **word type** for unique word forms, **word** for individual occurrences of a word type, **vocabulary type** for subword units in the tokenizer, and **token** for individual occurrences of those vocabulary types. We will also revise Equation 2 as **Average per-word loss** to reflect these definitions. Your interpretation of the word-level per-vocabulary loss calculation is correct: in the updated manuscript, Equation 2—renamed **Average per-word loss**—sums the cross-entropy losses of each word’s constituent tokens, and Figure 2 reports the mean of these per-word losses across the 2,500 most frequent words in FineWeb-Edu.
>
> ---
>
> > *3."Why embednorm does not reflect token frequency imbalance during training? + please explain this clearly in your manuscript"*
>
> Throughout training, both input and output embeddings reflect token-frequency imbalances, so frequent tokens develop larger embedding norms and thus produce higher logits. Under standard cross-entropy training, the gradient for the output embedding row of a target token $t$ is $\frac{\partial \mathcal{L}}{\partial E_{\mathrm{out},t}} = (p_t -1)h$ and  $\frac{\partial \mathcal{L}}{\partial E_{\mathrm{out},j}} = p_j h\ (j\neq t)$, respectively. $p_t$ is very small at initialization, the gradient of the target token is approximately $-h$, whereas non-target rows get $p_j h$ closer to 0. Consequently, each occurrence of a frequent token pulls its embedding by roughly $\lVert h\rVert$ along the hidden state direction, causing its norm to grow in proportion to its count, while rare tokens accrue only negligible norm increases. Although $\ell_{2}$ weight decay can slow this norm inflation for frequent tokens, its uniform shrinkage causes rare-token embedding norm, which receives negligible positive gradients, to shrink near zero [6]. By constraining a fixed unit $\ell_{2}$ norm on every embedding row on every step during training, we prevent embedding norms from encoding token-frequency imbalance during training. We have included this explanation in Appendix D, but will present this more clearly in the rest of the paper. Thank you for your guidance.
>
> ---
>
> > *4."What artifacts specifically will be released?"*
>
> We will release a 24K-196K tokenizer trained on Fineweb-edu (10B gpt2 tokens sample), openwebtext, and tokenized text, respectively. We will also release base and embednorm models trained on fineweb-edu (5 different seeds from 24K-196K vocab size).
>
> ---
>
> > *5."cite Magnusson et al 2024 in section 3.5 properly. what’s section 3.5 novelty against Magnusson et al 2024?"*
>
> We appreciate your correction and will incorporate your feedback. While [1] first identified that global loss is dominated by common tokens, our study further demonstrates that increasing the vocabulary further amplifies this dominance. Our work explains that a larger vocabulary reduces the complexity of the tokenized text and makes it more compressible by merging frequent non-iid character patterns into single tokens and boosting their relative frequency. This increased compressibility enables the model to learn and predict these frequent patterns more easily (even under an unigram assumption), which in turn lowers the cross‑entropy loss.
>
> ---
>
> > *6. "Why exactly bigger vocabularies increase certainty on the top tokens?"*
>
> Thank you for this insightful question. Expanding the vocabulary increases the relative frequency of common tokens as the total token count of the training corpus reduces. Even if frequent character strings are segmented as a single token (Figure 1b), expanding the vocabulary size further decreases the overall token count, which keeps increasing the relative frequency of common tokens. This amplified token‑frequency imbalance facilitates the language model’s ability to learn and predict these frequent tokens, as they occur more often with larger vocabularies. We provided a detailed analysis in Appendix A.
>
> ---
>
> > *7. "line 34: what is meant by “marginal ones” and mistakes on “them”?"*
>
> The marginal probabilities represent each token’s relative frequency, and “mistakes on them” indicates mistakes on rare tokens. Because the conditional probability of a rare token is very low, mispredicting such tokens incurs a large penalty.
>
> ---
>
> > *8. "Explain clearly what is the definition of segmentation efficiency. What do you want to show in figure 1c?"*
>
> We apologize for the previous ambiguity in our definition of segmentation efficiency. Segmentation efficiency gauges the average token count per word, that is, how many tokens the tokenizer needs, on average, to encode each of the 2,500 most frequent words. In Figure 1c, we aimed to illustrate that the frequent words are highly overlapping across datasets by showing that the overlap between the top N words of fineweb-edu and openwebtext remains consistently high. In response to the reviewer’s suggestion, we will expand the y‑axis range for greater clarity. Thank you for your advice.
>
> ---
>
> > *9. "What frequent words have you used in figure 2?"*
>
> We use frequent 2,500 words in the fineweb-edu.
>
> ---
>
> > *10. "define frequent words coverage in the caption."*
>
> We define “frequent‑words coverage” as the proportion of the model’s total loss attributable to the tokens comprising the 2,500 most frequent words in the FineWeb‑Edu corpus.
>
> ---
>
> > *11. "Maybe just plot the loss on infrequent words in figure 2b?"*
>
> In Figure 2, we will present the average per-vocab loss for infrequent tokens for each tokenizer instead of rare words, since rare words vary across datasets and most of them are not represented by a single token.
>
> ---
>
> > *12. "What is the shaded area showing here? STD over all strings?"*
>
> Shaded area showing 5 different runs on different random seeds
>
> ---
>
> > *13. "Why use a box plot and why average per vocab loss is so wide here?"*
>
> We employ box plots to depict the variability in loss for the 2,500 most frequent words: the shaded box spans one standard deviation above and below the mean, and the central black line corresponds exactly to the average per‑word loss reported in Figure 2a. To enhance interpretability, we will substitute these box plots with a line graph.
>
> ---
>
> > *14. "Frequent word loss of embednorm decreases as vocab scales up? Hard to see in box plots"*
>
> For clarity, the following table reports the exact average frequent word loss for both the baseline model and the embedding‑norm variant.
>
> **Table A**
>
> | Model      | 24K   | 49K   | 98K   | 196K  |
> |------------|-------:|-------:|-------:|-------:|
> | Base       | 4.074  | 4.031  | 4.001  | 3.977  |
> | EmbedNorm  | 4.205  | 4.194  | 4.183  | 4.179  |
>
> ---
>
> > *15. "why you use a common crawl dump? Isn’t it enough to show common token overlap between Fineweb-edu and downstream benchmarks?"*
>
> We want to verify on an independent validation set that expanding the vocabulary leads to reduced cross‑entropy loss by enabling the model to better capture high‑frequency words. Moreover, empirical findings in [2] report a strong negative Pearson correlation (–0.93) between CC‑Main‑2023‑40 cross‑entropy loss and reasoning benchmark performance, further motivating its use as a predictive validation set.
>
> ---
>
> > *16. "again the caption refers to loss on infrequent tokens but doesn’t show it"*
>
> We will revise Figure 5 to match the presentation style of Figure 2. Following your recommendation, Figure 5a will illustrate the average loss for infrequent tokens.
>
> ---
>
> > *17. "Do you have any explanation on why larger models should lower loss more on common tokens? "*
>
> It is also an intriguing research direction to understand why larger models lower loss on common tokens. We hypothesize that this effect derives from **simplicity bias** [3,4,5]: the inductive bias of deep networks to favor lower complexity solutions, and the bigger the model, the stronger the bias. [3] demonstrates that larger GPT‑3 models produce output text with lower complexity, which mirrors that of natural data. Moreover, [4] shows that simplicity bias increases with depth, and [5] demonstrates that the representation of models converges as model size grows. Given a training corpus that is both sufficiently large and diverse, simplicity biases enable larger models to more effectively capture the shared, high‑frequency patterns across natural text, thereby achieving lower cross‑entropy loss.
>
> ---
> ### Concluding Remarks
> Thank you again for your thoughtful and detailed feedback. Your comments have helped us clarify key methodological details, tighten our terminology, and strengthen both our analyses and visualizations. Should any further questions arise or additional experiments be desired, we would be delighted to address them. Your insights have greatly improved our paper, and we look forward to submitting a stronger revision.
>
>
> ---
>
> ### References
>
> [1] “Paloma:  A Benchmark for Evaluating Language Model Fit”  Neurips 2024 (benchmark)
>
> [2] “compression represents intelligence Linearly” COLM 2024
>
> [3] “The No Free Lunch Theorem, Kolmogorov Complexity, and the Role of Inductive Biases in Machine Learning” ICML 2024
>
> [4] “The Low-Rank Simplicity Bias in Deep Networks” TMLR 2023
>
> [5] “The Platonic Representation Hypothesis” ICML 2024
>
> [6] “Fishing for Magikarp: Automatically Detecting Under-trained Tokens in Large Language Models” EMNLP 2024

---

> > ### Comment · Reviewer_5AqE · 2025-08-06
> > **Thank you for response**
> >
> > Thank you for the very extensive response. I'll maintain my position that the paper should be accepted. My only issues are with the presentation and clarity of the findings and I'm satisfied with how the authors have engaged with my feedback in that respect.

---

### Official Review · Reviewer_8nWv · 2025-07-08

**Clarity:** 3
**Significance:** 2
**Originality:** 2
**Rating:** 4
**Confidence:** 4

**Summary:**

This work studies the  impact of vocabulary  size in the tokenization of large language models. The authors fix all hyperparameters but the vocabulary size, and  train models with varying vocabulary size to study various metrics. They examine total loss, per vocabulary loss, and metrics related to tokenization (overlap, segment efficiency) at various scales.  From these experiments, the authors claim that a larger vocabulary size leads to the most common tokens contributing to a majority of loss, and so if the model can learn to be more confident on those the overall loss will be decreased. The provide further evidence that it is these most common tokens that matter most, the authors train a model where with the input and prediction layers constrained to have  unit norm per token, and show that the benefits provided  to the common tokens no longer appears.

**Questions:**

- Did the authors try  optimizing any other hyperparaemters when scaling  the vocabulary and the model (as in Section 4.3)? Relating to my comments in the above  section, especially when the model is scaled I doubt using the same learning rate is optimal.
- Is there a motivation for using the divergence from a uniform distribution to characterize heavy tailed-ness? My intuition is that while  a heavy tailed distribution would have a large divergence, there are likely many other distributions  that do as well which would not  be traditionally described as heavy tailed.
- Could the authors further comment on the relevance of the section on Renyi  entropy? I’m unsure how well it  connects  to the rest of the paper.

**Ethical Concerns:**

["NO or VERY MINOR ethics concerns only"]

**Final Justification:**

In my initial review I expressed some concern about sensitivity to hyperparemters, difficulty making comparsions across vocabulary  sizes,  and the impact of constraints on optimizaiton. The authors added some additional experimentation during the rebuttle period to try to address some of these concerns,  which I found helpful for justifying their  claims. While I still feel the results  could be stronger with some better  desigend experiments, I am comfortable with raising my score to a weak accept.

**Limitations:**

The authors address some limitations.

**Quality:**

2

**Strengths And Weaknesses:**

The paper is looking into an interesting question from  an angle that isn’t so common these days. Typically papers just assume scale is better without looking into why that actually is, so I appreciate the general  investigative direction. The paper is written in a very structured way where  questions are stated and answered  in a clear way.

The main  weakness I see in this paper is that while it’s  conclusions are possible, I feel that the experimentation lacks the rigour to  be  confident the the authors conclusion holds. Their central  claim is that increasing the vocabulary  size essentially makes the most common token classes more dominant. The mechanism this occurs by is that token merges create new, small, classes, with the number of classes growing faster than the  number of tokens in most dominant classes is diminished by merges. The verification of this hypothesis does have a few issues. First, it isn’t really  possible to  hold the model constant  and  change the vocabulary size, as this impacts the number of the parameters in  the embedding and prediction layer. While this could potentially increase the expressivity of the model, a second issue it creates is changing the  optimization dynamics. In  particular, the  authors never change the learning rate in their experiments, and changing the scale of the model can certainly impact what an optimal learning rate  would be. A very related example of this can be seen in  [1], where it is argued that low frequency and high frequency token classes have very different characteristics during optimization. It could be the case that with an appropriate learning rate, both the loss on  the common  and uncommon  tokens goes down, contrary to this work’s findings. 	It is also not exactly surprising that adding constraints to the embeddings increases loss, adding any constraint on an unconstrained optimization   problem would likely do so, I suspect the loss also increases for  the  least frequent classes under this constraint. In order to have  a stronger argument for the author's hyothesis, it would likely  be necessary to design more rigorious experiments to  better demonstrate a causal relationship, I feel in this case a causal relationship has been claimed somewhat prematurely.

A more minor comment, at times the the words “word” and  “token” are used interchangeable, which as the authors also point out those are not  the same things. I am also somewhat confused by the inclusion of Section 5, it seems somewhat out of place in the context of the rest of the paper.

[1]
"Heavy-Tailed Class Imbalance and Why Adam Outperforms Gradient Descent on Language Models"
NeurIPS (Spotlight), 2024
https://proceedings.neurips.cc/paper_files/paper/2024/file/350e718ff74062b4bac2c6ffd9e1ac66-Paper-Conference.pdf

---

> ### Author Rebuttal · Authors · 2025-07-31
>
> Dear reviewer 8nWv,
>
> We sincerely appreciate your time and effort. We respond to your comment in what follows:
>
> ---
> > *1. "Altering vocabulary axis invariably changes the embedding and prediction-layer parameters, thus it may enhance model expressivity."*
>
> In response to the reviewer’s comment, we conducted a controlled experiment comparing a 24K vocabulary model (122 M total parameters) with a 49K vocabulary variant of comparable size (124 M parameters) to ascertain whether the benefits of vocabulary enlargement arise primarily from increased embedding capacity. To isolate this effect, we reduced the hidden dimension from 768 to 648 and the feed‑forward dimension from 2048 to 1728, while maintaining identical model depth which known to have a dominant impact on performance [2,3]. As shown in the table below, the 49K model incurs higher cross‑entropy loss and delivers inferior results on downstream benchmarks (ARC‑E, HellaSWAG, PIQA, and SCIQ) compared to its 24K counterpart, indicating that simply expanding the embedding layer contributes far less to model expressivity than is often assumed.
>
> **Table A**
>
> | Vocab (Params) | CE Loss | Downstream (%) |
> |---------------:|--------:|---------------:|
> | 24K (122M)     |   3.179 |          54.73 |
> | 49K (124M)     |   3.563 |          50.68 |
> | 49K (161M)     |   3.171 |          55.44 |
>
>
> Additionally, we compare ce-loss of base model and tied embedding variants. Were increased embedding capacity the primary driver of loss reduction, the tied‑embedding model would be expected to incur substantially higher loss. However, as shown in the table below, the difference in cross‑entropy loss between these two configurations is negligible.
>
>
> **Table B**
>
> | CE Loss        |  24K  |  49K  |  98K  | 196K  |
> |:--------------:|:-----:|:-----:|:-----:|:-----:|
> | Base           | 3.179 | 3.163 | 3.147 | 3.136 |
> | Tied Embedding | 3.177 | 3.160 | 3.146 | 3.134 |
>
> ---
>
> > *2."6e-4 is an optimal learning rate for every experiments (24K-196K)?"*
>
> We selected a learning rate of 6e-4 because it was employed for GPT‑3 125M models [5]. To address the reviewer’s question, we conducted a learning‑rate sweep using four rates
> (2.4e-3, 1.2e-3, 1.5e-4, 7.5e-5). The table below reports the resulting cross‑entropy losses and word-level average per vocab loss (we’ll rename this as average per word loss) of frequent 2500 words in fineweb-edu across different vocabulary sizes and learning rates, using the same validation dataset described in Section 3.1. Although 1.2e-3 looks more optimal learning rate, same loss curve observed across different vocabulary size.
>
> **Table C**
>
>
> | CE Loss | 2.4e-3 | 1.2e-3 | 6.0e-4 | 1.5e-4 | 7.5e-5 |
> |:-------:|-------:|-------:|-------:|-------:|-------:|
> | 24K     |  3.156 |  3.141 |  3.179 |  3.484 |  3.779 |
> | 49K     |  3.135 |  3.114 |  3.163 |  3.461 |  3.751 |
> | 98K     |  3.122 |  3.099 |  3.147 |  3.443 |  3.736 |
> | 196K    |  3.111 |  3.086 |  3.136 |  3.429 |  3.725 |
>
>
> **Table D**
>
> | Freq. Word Loss | 2.4e-3 | 1.2e-3 | 6.0e-4 | 1.5e-4 | 7.5e-5 |
> |:---------------|-------:|-------:|-------:|-------:|-------:|
> | 24K            |  4.033 |  4.012 |  4.074 |  4.431 |  4.783 |
> | 49K            |  3.990 |  3.970 |  4.031 |  4.399 |  4.749 |
> | 98K            |  3.963 |  3.939 |  4.001 |  4.373 |  4.725 |
> | 196K           |  3.942 |  3.916 |  3.977 |  4.352 |  4.703 |
>
>
> ---
>
> > *3."Constraining the embeddings would also significantly increase rare token loss."*
>
> To address the reviewer’s concern, we conducted an additional experiment to evaluate the effect of embedding‑norm constraints on cross‑entropy loss for both frequent and infrequent tokens. Specifically, we computed the average per‑token loss for the 2,500 most frequent and 2,500 least frequent tokens using 5B character subset of fineweb-edu (our validation dataset, see section 3.1). Gap quantifies the loss difference between the base models and embedding-normalized models.  While the norm constraint does raise loss on rare tokens, we find that the primary driver of the overall increase in cross‑entropy loss is the elevated loss on frequent tokens. This effect occurs because constraining the embedding norm prevents the model from encoding token‑frequency imbalance during training. We provided a detailed explanation in Appendix D. We are grateful to the reviewer for prompting this clarification.
>
> **Table E**
>
> | Vocab Size | Model     | Avg Freq. Loss | Avg Rare Loss |
> |:-----------|:----------|--------------:|--------------:|
> | 24K        | base      |        4.0298 |        4.5019 |
> | 24K        | embednorm |        4.0690 |        4.5094 |
> | 24K        | Gap       |        0.0392 |        0.0075 |
>
> **Table F**
>
> | Vocab Size | Model     | Avg Freq. Loss | Avg Rare Loss |
> |:-----------|:----------|--------------:|--------------:|
> | 49K        | base      |        3.9942 |        4.9424 |
> | 49K        | embednorm |        4.0353 |        4.9438 |
> | 49K        | Gap       |        0.0411 |        0.0014 |
>
> **Table G**
>
> | Vocab Size | Model     | Avg Freq. Loss | Avg Rare Loss |
> |:-----------|:----------|--------------:|--------------:|
> | 98K        | base      |        3.9532 |        5.4145 |
> | 98K        | embednorm |        3.9988 |        5.4210 |
> | 98K        | Gap       |        0.0456 |        0.0065 |
>
> **Table H**
>
> | Vocab Size | Model     | Avg Freq. Loss | Avg Rare Loss |
> |:-----------|:----------|--------------:|--------------:|
> | 196K       | base      |        3.9369 |        6.2493 |
> | 196K       | embednorm |        3.9817 |        6.2581 |
> | 196K       | Gap       |        0.0448 |        0.0088 |
>
>
> ---
>
> > *4."“word” and “token” are used interchangeable,which as the authors also point out those are not the same things."*
>
> We apologize for having used “word” and “token” interchangeably throughout the manuscript. In the revised version, we will distinguish and apply these terms consistently. Thank you for your feedback.
>
> ---
>
> > *5."Why use JSD for measuring heavy-tailedness of distribution?"*
>
> We use the Jensen–Shannon divergence from a uniform distribution to quantify token frequency imbalance and thereby evaluate the compressibility (or complexity) of the tokenized text alongside Shannon entropy. This work focuses exclusively on token‑frequency imbalance, not on the distribution’s tail shape; references to “heavy‑tailedness” were unintended. In particular, we will remove the phrase “leading to a heavier long‑tail distribution” from Figure 1’s caption. Thank you for drawing our attention to this issue.
>
> ---
>
> > *6."How section 5 (discussion) connects with rest of paper"*
>
> Section 5 addresses the apparent discrepancy with prior work [4], which finds that increased token‑frequency imbalance degrades machine translation performance. We wanted to explain what leads to different conclusion in section 5. In machine translation, source and target vocabularies have minimal overlap—terms frequent in English rarely appear in French and vice versa—whereas in monolingual pre‑training, the training corpus and evaluation benchmarks share extensive lexical overlap, including many common high‑frequency tokens. We wanted to point out that in machine translation, gains from reducing frequent token loss in the source language during training do not necessarily translate into improved performance in the target language and this fundamental contrast accounts for the somewhat different conclusions of [4] and our study.
>
> ---
>
> ### Concluding Remarks
> Your feedback has been instrumental in this process, and we sincerely extend our gratitude for your invaluable insights. Should you have any inquiries or require clarifications about our rebuttal, please don't hesitate to reach out. We are eager to address any concerns and elucidate potential ambiguities in greater depth.
>
> ---
> ### Reference
>
> [1] “Over-Tokenized Transformer: Vocabulary is Generally Worth Scaling” ICML 2025
>
> [2] “SCALE EFFICIENTLY: INSIGHTS FROM PRE-TRAINING AND FINE-TUNING TRANSFORMERS” ICLR 2022
>
> [3] “The Impact of Depth and Width on Transformer Language Model Generalization” Arxiv
>
> [4] “Tokenization and the Noiseless Channel” ACL 2023
>
> [5] “Language Models are Few-Shot Learners”

---

> > ### Comment · Reviewer_8nWv · 2025-08-05
> >
> > I thank the reviwers for their detailed response to my concerns. I am somewhat more comfortable with the conclusions of this work and I encourage the authors to add any additional experiments into further revisions of the paper. I will  raise my score to a weak accept given these additional  results.

---

### Author Response · Authors · 2025-08-05

Dear Reviewers,

Thank you again for your thoughtful time and feedback on our submission.

We have submitted detailed responses to your comments and would greatly appreciate it if you could take a moment to review them when convenient. If there are any questions or further points that need clarification, we are more than happy to engage in discussion.

We look forward to your continued guidance.

Best regards,

---

### Note · Authors · 2025-08-12

Dear Area Chair and Reviewers,

We sincerely appreciate your time and service to the NeurIPS community. Our paper offers a controlled study that isolates vocabulary size as the only independent variable (5NL9), thereby clarifying why increasing vocabulary can benefit pre-training (8nWv, Md1b). We pair this theoretical motivation with carefully controlled experiments (8nWv, Md1b, 5AqE), yielding insights that are directly useful for language-model practitioners (5AqE).

In light of our discussion with the reviewers, we believe the rebuttal has successfully addressed their concerns, and we will incorporate their suggestions and comments to further strengthen the manuscript. We are again very thankful to the Area Chairs for their hard work and to the reviewers for their careful, thoughtful, and thorough feedback.

Sincerely,

The Authors

---

### Decision · Program_Chairs · 2025-09-17

**Decision:**

Accept (poster)

**Comment:**

The paper investigates the role of vocabulary size in LLMs in a controlled setting, and identifies "sharper frequency imbalance" as a key factor behind the benefits of larger vocabularies: even when the most frequent words remain their own tokens, increasing vocab size makes the rarer tokens less frequent in the data, thus making it easier to learn the more frequent tokens more efficiently, leading to overall improvements in learning efficiency. These are interesting findings, and the experiments were carried carefully. Overall, the reviewers were positive about the paper, and I recommend acceptance.

Please make sure to address the presentation issues raised by reviewer 5AqE in the final revision, and please do release the artifacts mentioned.